# Minimax Adaptive Online Nonparametric Regression over Besov spaces

**Paul Liautaud**
Sorbonne Université, CNRS, LPSM
F-75005 Paris, France
`paul.liautaud@sorbonne-universite.fr`

**Pierre Gaillard**
Université Grenoble Alpes, Inria
CNRS, Grenoble INP, LJK
38000 Grenoble, France
`pierre.gaillard@inria.fr`

**Olivier Wintenberger**
Sorbonne Université, CNRS, LPSM
F-75005 Paris, France
Institut Pauli, CNRS and University of Vienna
Oskar-Morgenstern-Platz 1, 1090 Wien, Austria
`olivier.wintenberger@sorbonne-universite.fr`

## Abstract

We study online adversarial regression with convex losses against a rich class of continuous yet highly irregular competitor functions, modeled by Besov spaces $B_{pq}^s$ with general parameters $1 \leqslant p, q \leqslant \infty$ and smoothness $s > \frac{d}{p}$. We introduce an adaptive wavelet-based algorithm that performs sequential prediction without prior knowledge of $(s, p, q)$, and establish minimax-optimal regret bounds against any comparator in $B_{pq}^s$. We further design a locally adaptive extension capable of sequentially adapting to spatially inhomogeneous smoothness. This adaptive mechanism adjusts the resolution of the predictions over both time and space, yielding refined regret bounds in terms of local regularity. Consequently, in heterogeneous environments, our adaptive guarantees can significantly surpass those obtained by standard global methods.

## 1 Introduction

We consider the online regression framework [5, 6], where inputs $x_1, \ldots, x_t, \ldots \in \mathcal{X}$ arrive in a stream, and the task is to sequentially predict a response $\hat{f}_t(x_t) \in \mathbb{R}$ using an online predictive algorithm $\hat{f}_t : \mathcal{X} \to \mathbb{R}$ based on past observations $s = 1, \ldots, t - 1$ and the current input $x_t$. The goal is to design a sequence of predictors $(\hat{f}_t)$ in the *competitive approach*, i.e., with guarantees that hold uniformly over all individual (and potentially adversarial) data sequences. Prediction accuracy is assessed over time using a sequence of convex loss functions $(\ell_t)_{t \geqslant 1}$, for instance $\ell_t(\hat{f}_t(x_t)) = |\hat{f}_t(x_t) - y_t|$ or $(\hat{f}_t(x_t) - y_t)^2$, where $y_t$ is the observed response associated with $x_t$. After $T \geqslant 1$ rounds, the performance of the algorithm is measured through its *regret* with respect to competitive continuous functions $f$,

$$R_T(f) := \sum_{t=1}^{T} \ell_t(\hat{f}_t(x_t)) - \sum_{t=1}^{T} \ell_t(f(x_t)). \tag{1}$$

Much of the early literature [7, 17, 23, 25] focuses on competitors $f$ belonging to smooth benchmark classes, such as Lipschitz or kernel-based functions. In this work, we extend this setting by designing constructive algorithms that are competitive with a much richer class of prediction rules, namely functions in general Besov spaces [20, 37, 40]. Building on wavelet-based representations, we design an adaptive algorithm that achieves optimal regret performance (1), against broad classes of prediction rules, modeled by Besov spaces. Wavelets [8, 11] are indeed a powerful and widely used tool for

39th Conference on Neural Information Processing Systems (NeurIPS 2025).

capturing local features and regularities in signals. Their applications range from image segmentation and change point detection to EEG analysis and financial time series. Moreover, in many practical scenarios, the environment may exhibit spatial heterogeneity, with varying degrees of regularity across the domain. This motivates the need for methods that can adapt locally to different smoothness levels. To tackle this challenge, we develop a locally adaptive algorithm [28, 29] that sequentially adjusts the resolution of its predictions over space, effectively adapting to inhomogeneous regularity. Our analysis provides minimax optimal regret guarantees that depend on the local smoothness of the target function, improving upon globally-tuned algorithms.

**Wavelets and multiscale approaches.** Classical wavelet-based methods for statistical function estimation have been primarily developed and analyzed in the batch (i.i.d.) setting, where the entire dataset is available upfront. Notable examples include the wavelet shrinkage procedure of [15], which achieves near-minimax estimation rates over Besov spaces. More generally, wavelets play a central role in the signal processing and compressed sensing framework developed by [30], where they are well understood and widely applied. In the context of adaptive and nonparametric estimation, [1, 2] introduced universal algorithms based on tree-structured approximations, closely related in spirit to wavelet thresholding. While these methods are computationally efficient and amenable to online implementation, their theoretical analysis is performed in the batch statistical learning setting and focuses on specific classes of approximation spaces. Multiscale and chaining ideas have also emerged in the online learning literature, beginning with the early work of [5] and continuing more recently in [34] (in a non-constructive fashion) and [17], although typically without relying on explicit wavelet constructions. Recently, [43] studied an online algorithm that combines discrete wavelets with parameter-free learning to minimize dynamic regret under general convex losses. Beyond this, the combination of wavelet-based representations with principled online nonparametric learning guarantees remains largely unexplored. Our work contributes to this direction by developing an online algorithm that leverages multiscale wavelet structures with theoretical regret guarantees over large nonparametric Besov function classes.

**Online nonparametric regression.** A classical line of work in online regression focuses on competing with smooth benchmark functions with a given degree of smoothness $s > 0$. For instance, [7, 17, 29] design constructive online algorithms that achieve optimal regret against Lipschitz functions ($s \leqslant 1$) by using chaining-based techniques and exploiting regularity properties such as uniform continuity to build refined predictors. Much of the early literature also focused on reproducing kernel Hilbert spaces (RKHS) [3, 4, 19, 25, 38], which correspond to the case where the smoothness index satisfies $s > \frac{d}{2}$ and $p = 2$. This setting offers convenient geometric properties, such as inner products and representer theorems, but it excludes many natural function classes of interest, e.g. general $L^p(\mathcal{X})$ spaces with $p \geqslant 1$, Sobolev spaces with low smoothness $s$, or more generally Besov spaces. A key milestone in the direction of generalizing beyond RKHS is the work [40], which introduces the method of defensive forecasting to compete with *wild prediction rules*, i.e., rules drawn from general Banach spaces (e.g., $L^p(\mathcal{X}), p \geqslant 2$). Their framework shows that online learning is possible in highly irregular settings and provides regret bounds that depend on the geometry of the underlying Banach space. However, their analysis yields bounds that depend solely on the integrability parameter $p \geqslant 2$, and does not account for any additional smoothness structure that the benchmark functions may possess. This motivates the need for online learning strategies that adapt not only to *integrability*, but also to spatial *regularity* or *smoothness*. Another paper in this line is [42], where they study the performance of Sobolev kernels on restricted classes of Sobolev spaces $W_p^s(\mathcal{X})$ with integrability $p \geqslant 2$ and smoothness $s > \frac{d}{p}$. Going one step further, our paper proposes an algorithm with regret guarantees against any competitor in general Besov spaces $B_{pq}^s$, for any integrability parameters $1 \leqslant p, q \leqslant \infty$ and smoothness $s > \frac{d}{p}$. This generalizes and improves upon previous methods by addressing a broad range of function spaces. More importantly, none of the constructive methods mentioned above provide the minimax optimal rate for generic Besov spaces, as established by [34]. To the best of our knowledge, we present the first constructive and adaptive algorithm that bridges wavelet theory with online nonparametric learning, while providing minimax optimal regret guarantees against functions in general Besov spaces. Table 1 summarizes our contributions and the corresponding regret rates in the literature.

**Local adaptivity in inhomogeneous smoothness regimes.** Many real-world functions exhibit spatially varying regularity, motivating the development of locally adaptive methods. In the batch setting, [14] pioneered spatially adaptive wavelet estimators that adjust to unknown smoothness. Bayesian approaches such as [36] further model locally Hölder functions with hierarchical priors. In a

**Table 1:** Comparison of regret rates and parameter requirements for online regression.

| Paper | Setting ($(\ell_t)$ square losses, $s > \frac{d}{p}$) | Input Parameters | Regret Rate |
|---|---|---|---|
| Vovk [39] | $f \in B^s_{pq}, p, q \geqslant 1$ | $s, p, B \geqslant \|f\|_\infty$ | $T^{1-\frac{s}{s+d}}$ |
| Vovk [40] | $f \in B^s_{pq},\ p \geqslant 2, q \in [\frac{p}{p-1}, p]$ | $s, p, B \geqslant \|f\|_\infty$ | $T^{1-\frac{1}{p}}$ |
|  | $f \in W^s_\infty = \mathscr{C}^s,\ p = \infty, s \in [\frac{d}{2}, 1]$ |  | $T^{1-\frac{s}{d}+\varepsilon}$ |
| Gaillard and Gerchinovitz [17] | $f \in W^s_p,\ p \geqslant 2,\ s \geqslant \frac{d}{2}$ | $s, p, B \geqslant \|f\|_\infty$ | $T^{1-\frac{2s}{2s+d}}$ |
|  | $f \in W^s_p,\ p > 2,\ s < \frac{d}{2}$ |  | $T^{1-\frac{s}{d}}$ |
| Zadorozhnyi et al. [42] | $f \in W^s_p,\ p \geqslant 2,\ s \geqslant \frac{d}{2}$ | $s, p$ | $T^{1-\frac{2s}{2s+d}+\varepsilon}$ |
|  | $f \in W^s_p,\ p > 2,\ s < \frac{d}{2}$ |  | $T^{1-\frac{s}{d}\frac{p-d/s}{p-2}+\varepsilon}$ |
| **This work** - Alg. 2 | $f \in B^s_{pq}, p, q \geqslant 1, s \geqslant \frac{d}{2}$ or $p \leqslant 2$ | $S \geqslant s, \varepsilon < s - \frac{d}{p}, B \geqslant \|f\|_\infty$ | $T^{1-\frac{2s}{2s+d}}$ |
|  | $f \in B^s_{pq}, p > 2, q \geqslant 1, s < \frac{d}{2}$ |  | $T^{1-\frac{s}{d}}$ |

distribution-free framework, [21, 27] introduced the notion of *average smoothness*, based on averaging local Hölder semi-norms at a fixed degree of regularity. In the online setting, [28, 29] developed algorithms that sequentially adapt to local smoothness across time. However, these approaches typically focus on adapting to local norms while assuming a fixed degree of regularity. In contrast, our method jointly adapts to both the local regularity norm and the local smoothness exponent, enabling a data-driven compromise that is well suited to highly inhomogeneous environments.

**Context and notation.** Throughout the paper, we assume the following. $\mathcal{X}$ denotes a compact domain of $\mathbb{R}^d$, $d \geqslant 1$. For any subset $\mathcal{X}' \subseteq \mathcal{X}$, we set its diameter as $|\mathcal{X}'| = \sup_{x,y \in \mathcal{X}'} \|x - y\|_\infty$. Without loss of generality, we assume that $\mathcal{X}$ is a regular hypercube of volume $|\mathcal{X}|^d$. We denote the horizon of time by $T \geqslant 1$. The sequence losses $(\ell_t)$ are assumed to be convex and $G$-Lipschitz for some $G > 0$. For any natural integer $k \in \mathbb{N}$, we denote $[k] := \{0, \ldots, k\}$.

## 2 Background and function representation

We consider compactly supported functions $f : \mathcal{X} \to \mathbb{R}$ that lie in $L^2(\mathcal{X})$ equipped with the standard inner product $\langle f, g \rangle = \int_{\mathcal{X}} f(x)g(x)\,dx$. To design a sequential algorithm we rely on a multiscale representation of $f$ based on an orthonormal wavelet basis. For a chosen starting scale $j_0 \in \mathbb{N}$, we write:

$$f = \sum_{k \in \bar{\Lambda}_{j_0}} \alpha_{j_0,k} \phi_{j_0,k} + \sum_{j=j_0}^{\infty} \sum_{k \in \Lambda_j} \beta_{j,k} \psi_{j,k}, \tag{2}$$

where the families $(\phi_{j_0,k})_{k \in \bar{\Lambda}_{j_0}}$ and $(\psi_{j,k})_{k \in \Lambda_j, j \geqslant j_0}$ form an orthonormal basis of $L^2(\mathcal{X})$. We now highlight the key properties of the expansion (2), and refer the interested reader to Appendix E for further details.

**Scaling (coarse-scale) component.** The functions $\phi_{j_0,k}(x) := 2^{j_0 d/2} \phi(2^{j_0} x - k)$ are the *scaling functions* at resolution level $j_0$, constructed from a fixed *father wavelet* $\phi$. They span

$$V_{j_0} := \mathrm{span}\{\phi_{j_0,k} : k \in \bar{\Lambda}_{j_0}\},$$

where the index set $\bar{\Lambda}_{j_0}$ satisfies $|\bar{\Lambda}_{j_0}| \leqslant \lambda 2^{j_0 d}$ for some constant $\lambda > 0$. The corresponding coefficients $\alpha_{j_0,k} := \langle f, \phi_{j_0,k} \rangle$ are known as the *scaling coefficients*.

**Wavelet (detail-scale) components.** The functions $\psi_{j,k}(x) := 2^{jd/2} \psi(2^j x - k)$ are the *wavelet functions* at scale $j$, obtained from a fixed *mother wavelet* $\psi$. Here, the multi-index $k$ encodes both spatial position and directional information in $d$ dimensions - see Appendix E for a brief summary of the tensor-product construction used to define such wavelets in dimension $d \geqslant 1$. The detail space at level $j$ is defined as

$$W_j := \mathrm{span}\{\psi_{j,k} : k \in \Lambda_j\},$$

where $\Lambda_j$ indexes the active wavelet functions whose supports intersect $\mathcal{X}$, and $|\Lambda_j| \leqslant \lambda 2^{jd}$, $j \geqslant j_0$, for the same constant $\lambda > 0$ as above with no loss of generality. The coefficients $\beta_{j,k} := \langle f, \psi_{j,k} \rangle$ are called *detail coefficients* at scale $j$. Within the multiresolution analysis framework, the sequence of spaces $(V_j)_{j \in \mathbb{Z}}$ forms a nested hierarchy with $V_j \subset V_{j+1}$ and dense union in $L^2(\mathbb{R}^d)$, while the wavelet spaces $W_j$ are orthogonal complements such that $V_{j+1} = V_j \oplus W_j$. Figure 1 illustrates the hierarchical, stage-wise approximation process over levels $j$ that enables multiresolution analysis.

The decomposition (2) is a classical result from multiresolution analysis (see [22, Chap. 3] for a deeper introduction), and holds for all functions in $L^2(\mathcal{X})$ when the basis functions are derived from appropriately constructed wavelets. More generally, similar dyadic expansions can also be built from splines or piecewise polynomial systems. In this work, we focus on the wavelet setting as described above; we refer to [11, Chap. 6] and [8] for further details.

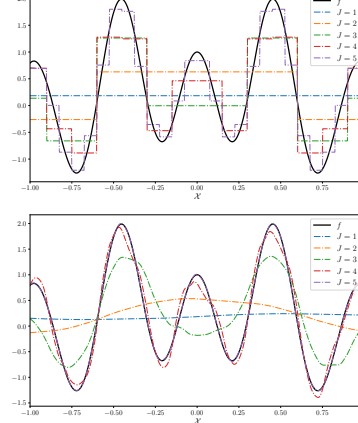

One property of this representation (2) is that it can begin at any arbitrary scale $j_0 \in \mathbb{N}$, offering flexibility to adapt the starting resolution level. In particular, we will later allow $j_0$ to be selected in a data-driven and local fashion.

Throughout the paper, we do not rely on a specific wavelet basis, but require that it satisfies the standard *S-regularity property* for some $S \in \mathbb{N}^*$, as recalled in Definition 2 in Appendix E. This condition ximplies compact support, smoothness, vanishing moments, and bounded overlap. Notable examples include the compactly supported orthonormal wavelets of Daubechies [11, Chap. 7], and the biorthogonal, symmetric, and highly regular wavelet bases of Cohen et al. [9] - see Figure 1 for an illustration of approximation with $S$-regular Daubechies wavelets. When

Figure 1: Approximation with Daubechies wavelets of regularity $S = 1$ (top) and $S = 5$ (bottom) at levels $J = 1, \ldots, 5$.

working with $S$-regular wavelet bases, the expansion in (2) converges not only in $L^2(\mathcal{X})$ but also in other function spaces, such as $L^p(\mathcal{X})$ for $p \geqslant 1$ (or the space of uniformly continuous functions) depending on whether $f \in L^p(\mathcal{X}), p \geqslant 1$. This broader convergence behavior is a key reason for adopting such bases.

**Approximation results with wavelets.** Approximation properties of wavelet expansions are by now classical and well understood; see, e.g., [8, 13, 20] for reviews. In particular, for functions $f$ belonging to smoothness spaces such as Besov spaces $B^s_{pq}$ with $s > \frac{d}{p}$ (so that $f \in L^\infty(\mathcal{X})$), one can construct an approximation $\hat{f}$ using *nonlinear methods* — see [12]. For instance, the so-called best $N$-term approximant $\hat{f}$ in a $S$-regular wavelet basis ($S > s$, see Definition 2) achieves the bound $\|f - \hat{f}\|_\infty \lesssim N^{-s/d}$, where the hidden factor depends on the wavelet basis and the norm of the target function $f$. The precise construction of $\hat{f}$ and justification of this rate are provided in the Appendix (see the proof of Theorem 1). These rates will serve as a benchmark in our analysis of the regret (1).

## 3 Parameter-free online wavelet decomposition

In this section, we develop a sequential algorithm that performs an online, parameter-free decomposition of incoming data using wavelets. The key idea is to learn wavelet coefficients incrementally - without prior knowledge of the function's regularity or the optimal resolution depth - while obtaining strong regret guarantees over broad function classes.

### 3.1 Algorithm: Online Wavelet Decomposition

Let $\{\phi_{j_0,k}, \psi_{j,k}\}$ denote an $S$-regular wavelet basis as introduced in the previous section. We consider an online predictor based on a wavelet expansion (2) that begins at scale $j_0 \in \mathbb{N}$ and is truncated at level $J \geqslant j_0$, where the predictor at time $t \geqslant 1$ takes the form:

$$\hat{f}_t(x) = \sum_{k \in \bar{\Lambda}_{j_0}} \alpha_{j_0,k,t} \phi_{j_0,k}(x) + \sum_{j=j_0}^{J} \sum_{k \in \Lambda_j} \beta_{j,k,t} \psi_{j,k}(x), \tag{3}$$

where the scaling and detail coefficients $\{\alpha_{j_0,k,t}, \beta_{j,k,t}\}$ are updated sequentially over time.

**Online optimization of wavelet coefficients.** Our algorithm maintains and updates the collection of scaling and detail coefficients $\{\alpha_{j_0,k,t}, \beta_{j,k,t}\}$ in (3) across scales $j$ and positions $k$. At each round $t \geqslant 1$, after observing a new input $x_t$, the prediction is computed using only the coefficients and basis functions whose support intersects $x_t$. This defines the active index set at time $t$:

$$\Gamma_t := \big\{(j_0, k) : \phi_{j_0,k}(x_t) \neq 0\big\} \cup \bigcup_{j=j_0}^{J} \big\{(j,k) : \psi_{j,k}(x_t) \neq 0\big\} \subsetneq \bar{\Lambda}_{j_0} \cup \bigcup_{j=j_0}^{J} \Lambda_j. \tag{4}$$

Only the coefficients indexed by $\Gamma_t$ are updated at time $t$, based on gradient feedback from the loss. The full procedure is summarized in Algorithm 1.

---

**Algorithm 1:** Online Wavelet Decomposition at time $t$

---

**Input** : Current coefficients $\{\alpha_{j_0,k,t}, \beta_{j,k,t}\}$; active index set $\Gamma_t$ defined in (4).

**1** Predict with coefficients in $\Gamma_t$

$$\hat{f}_t(x_t) = \sum_{(j,k) \in \Gamma_t} c_{j,k,t}\, \varphi_{j,k}(x_t)$$

where $\varphi_{j,k}$ stands for either $\phi_{j_0,k}$ or $\psi_{j,k}$; similarly, $c_{j,k,t} = \alpha_{j_0,k,t}$ or $\beta_{j,k,t}$;

**2** Receive gradients $\{g_{j,k,t}\}$ associated with the active coefficients, defined in Eq. (5);

**3** **for** $(j,k) \in \Gamma_t$ **do**

**4**     Update $c_{j,k,t}$ to $c_{j,k,t+1}$ by approximately minimizing

$$c \mapsto \ell_t\big(\hat{f}_t(x_t) - c_{j,k,t}\varphi_{j,k}(x_t) + c\,\varphi_{j,k}(x_t)\big)$$

    using corresponding gradient $g_{j,k,t}$ and a parameter-free update rule satisfying Assumption 1;

**Output** : Updated coefficients $\{\alpha_{j_0,k,t+1}, \beta_{j,k,t+1}\}$.

---

**Computation of gradients.** We assume that after making its prediction, the algorithm receives first-order feedback in the form of gradients $\{g_{j,k,t}\}$ with respect to each active coefficient $c_{j,k,t}$, where $c_{j,k,t}$ denotes either a scaling coefficient $\alpha_{j_0,k,t}$ or a wavelet coefficient $\beta_{j,k,t}$. These gradients are efficiently computed using the chain rule:

$$g_{j,k,t} = \Big[\nabla_c \ell_t\big(\hat{f}_t(x_t) - c_{j,k,t}\varphi_{j,k}(x_t) + c\,\varphi_{j,k}(x_t)\big)\Big]_{c=c_{j,k,t}} = \ell'_t(\hat{f}_t(x_t)) \cdot \varphi_{j,k}(x_t), \qquad (5)$$

where $\varphi_{j,k}$ is either $\phi_{j_0,k}$ or $\psi_{j,k}$ depending on the scale, and $\ell'_t$ is the derivative of the loss function with respect to its prediction argument. Note that the gradient expression in (5) vanishes whenever the corresponding basis function satisfies $\varphi_{j,k}(x_t) = 0$. This justifies restricting the optimization step at round $t$ to the active set $\Gamma_t$ defined in (4).

**Assumption on the gradient step.** To analyze the regret (1) of our method, we assume that the update rule used in Algorithm 1 satisfies a parameter-free regret guarantee of the following form.

**Assumption 1** (Parameter-free regret bound). *Let $T \geqslant 1$ and suppose $g_1, \ldots, g_T \in [-\hat{G}, \hat{G}]$ are the gradients observed over time. We assume that the coefficient update rule satisfies, for any $c \in \mathbb{R}$,*

$$\sum_{t=1}^{T} g_t(c_t - c) \leqslant |c - c_1| \Big( C_1 \sqrt{\sum_{t=1}^{T} |g_t|^2} + C_2 \hat{G} \Big)$$

*for some $C_1, C_2 > 0$.*

This regret bound holds with an additional term $\hat{G}\varepsilon$, $\varepsilon > 0$ as small as possible, for a broad class of first-order online learning algorithms, such as online mirror descent with self-tuned learning rates or coin-betting style updates [33]. For simplicity we consider that Assumption 1 holds omitting this additional term and the dependence in the hyperparameter $\varepsilon > 0$ in the sequel. These algorithms are referred to as "parameter-free", and they provide optimal adaptivity to $|c|$, at the expense of logarithmic factors absorbed in $C_1$ and $C_2$ (see, e.g., [10, 32]). Note also that this type of algorithms are explicit and maintain their iterates through a closed-form update, resulting in low computational complexity—see, for instance, the update rule in [32, Eq. (9)].

### 3.2   Regret analysis of Online Wavelet Decomposition (Alg. 1)

In this section, we analyze the regret performance of Algorithm 1 under general convex losses $(\ell_t)$, and against a broad class of potentially irregular prediction rules, specifically those lying in Besov spaces $B_{pq}^s(\mathcal{X})$, which we introduce later. We focus in particular on the case $s > \frac{d}{p}$, ensuring that the competing functions are continuous and bounded; see the standard embedding results in [20, 37]. As a corollary, we show that Algorithm 1 achieves minimax-optimal rates when competing against functions in Hölder spaces, while automatically adapting to the unknown regularity of the target function.

**Besov spaces.** Besov spaces $B_{pq}^s(\mathcal{X})$ constitute a classical family of function spaces indexed by three parameters: a smoothness parameter $s > 0$, an integrability parameter $p \in [1, \infty]$, and a summability parameter $q \in [1, \infty]$. For those un-familiar with Besov spaces, this space can be intuitively viewed as the space of functions with $s > 0$ derivatives in $L^p(\mathcal{X})$, with $p \geqslant 1$, and parameter $q \geqslant 1$ allows for additional finer control of the regularity of the underlying functions. These spaces interpolate between Sobolev and Hölder spaces and are designed to capture both smooth and non-smooth behaviors in functions. There exist several equivalent definitions of Besov spaces (e.g. using differences, or interpolation theory): we refer to [20, 22, 37] for detailed and general background on Besov spaces. In this work, we adopt the wavelet characterization, which is par-ticularly well suited for the analysis of our wavelet-based algorithm.

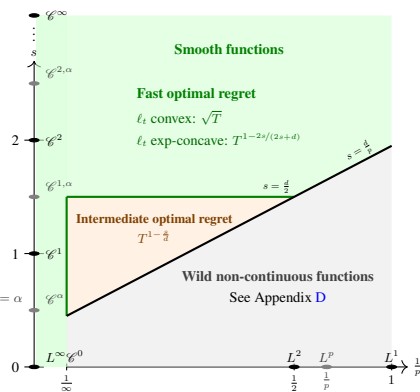

Figure 2: Diagram $(d/p, s)$ of regret regimes against Besov spaces.

Let $s > 0$ and let $\{\phi_{j_0,k}, \psi_{j,k}\}$ be an orthonormal $S$-regular wavelet basis with $S > s$ (see Definition 2). We say that a function $f$ belongs to the Besov space $B_{pq}^s$ if the following wavelet-based norm is finite, with $s' = s + \frac{d}{2} - \frac{d}{p}$:

$$\|f\|_{B_{pq}^s} := \|\boldsymbol{\alpha}_{j_0}\|_p + \left( \sum_{j \geqslant j_0} 2^{jqs'} \|\boldsymbol{\beta}_j\|_p^q \right)^{1/q}, \quad \text{if } 1 \leqslant q < \infty,$$

$$\|f\|_{B_p^s} := \|\boldsymbol{\alpha}_{j_0}\|_p + \sup_{j \geqslant j_0} 2^{js'} \|\boldsymbol{\beta}_j\|_p, \quad \text{if } q = \infty,$$

(6)

where $\boldsymbol{\alpha}_{j_0} = (\langle f, \phi_{j_0,k} \rangle)_{k \in \bar{\Lambda}_{j_0}}$ denotes the vector of scaling coefficients at level $j_0$, and $\boldsymbol{\beta}_j = (\langle f, \psi_{j,k} \rangle)_{k \in \Lambda_j}$ are the wavelet coefficients at scale $j \geqslant j_0$.

We now present our first result, establishing regret guarantees for Algorithm 1 when competing against irregular but bounded prediction rules.

**Theorem 1** (Regret against Besov predictors). *Let $T \geqslant 1$, $s > 0, 1 \leqslant p, q \leqslant \infty$ with $s - \frac{d}{p} > \varepsilon > 0$. Let $\{\phi_{j_0,k}, \psi_{j,k}\}$ be an $S$-regular wavelet basis (Definition 2) for some $S > s$. Suppose Algorithm 1 is run with updates satisfying Assumption 1, starting at $\boldsymbol{\alpha}_{j_0} = 0, \boldsymbol{\beta}_j = 0, j \geqslant j_0$ and using a wavelet expansion (3) from scale $j_0 = 0$ to $J = \lceil \frac{S}{d\varepsilon} \log_2 T \rceil$. Then, for any $f \in B_{pq}^s(\mathcal{X})$, the regret satisfies:*

$$R_T(f) \leqslant CG\|f\|_{B_{pq}^s} \begin{cases} \sqrt{T} & \text{if } s \geqslant \frac{d}{2} \text{ or } p < 2 \\ T^{1-s/d} & \text{else}, \end{cases}$$

*where $C = C(\lambda, \psi, s, p, C_1, C_2) > 0$ depend only on $s$, $p$, the wavelet basis, and the $C_1, C_2$ in Assumption 1.*

Theorem 1 is proved in Appendix A, where we provide the full statement including explicit constants. Our bounds are *minimax-optimal* in the regimes $s > \frac{d}{p}$ when facing convex losses; see [35]. Note that logarithmic factors in $T$ may be absorbed into the constant in the regret bound of Theorem 1, typically in the case $p < q$. The details are provided in Appendix A.

**Adaptivity and tradeoff.** Importantly, our procedure is *adaptive* to the Besov norm $\|f\|_{B_{pq}^s}$, and the parameters $(s, p, q)$ whenever $s < S$. Notably, via the usual embeddings, $f$ can belong simultaneously to several Besov spaces $B_{p\infty}^s$ with different norms $\|f\|_{B_{p\infty}^s}$ depending on $s$ and $p$. Remarkably, our algorithm effectively competes against any oracle associated to the best (which are not necessarily the largest) values of $(s, p)$, yielding a regret bound of type:

$$R_T(f) \leqslant \inf_{s,p} CG\|f\|_{B_{p\infty}^s} T^{r(s,p)} \tag{7}$$

where the infimum is taken over all admissible pairs $(s, p)$ such that $f \in B_{p\infty}^s(\mathcal{X})$, the exponent $r(s, p)$ reflects the rate in each regime according to $(s, p)$ (see Theorem 1), and the constant $C = C(\lambda, \psi, s, p, C_1, C_2)$ detailed in Appendix A.

**Complexity and choice of wavelet basis.** While one can use wavelets with infinite regularity (i.e., $S = \infty$), such as Meyer wavelets, these are not compactly supported in space and thus lack

good localization properties. In practice, it is more common to use compactly supported wavelets that offer a good trade-off between smoothness and spatial localization. Their compact support implies that most basis functions $\psi_{j,k}$ vanish at any given input $x_t$: only the indices $k \in \Lambda_j$ such that $x_t \in \mathrm{supp}(\psi_{j,k})$ contribute to the prediction. For example, Daubechies wavelets of regularity $S$ are supported on the hypercube $[0, S]^d$, so at most $O(S^d)$ coefficients per scale $j$ are nonzero at any point $x_t$. Among all wavelet families, the Haar basis (corresponding to $S = 1$) yields the most efficient updates, as its basis functions are non-overlapping, but it is limited to capturing piecewise constant (i.e., Lipschitz-1) regularity; see [29] for an algorithm exploiting this structure and Figure 1 for an illustration in the cases $S = 1$ and $S = 5$. Each wavelet coefficient is updated through a closed-form function of its scalar gradient, satisfying Assumption 1 and leading to $O(1)$ cost per coefficient and keeping the overall update as light as standard gradient descent (see [10, 32]). As a result, the per-round computational cost of our algorithm scales as $O(JS^d)$, where $J$ is the total number of levels.

**The case of Hölder function spaces $\mathscr{C}^s(\mathcal{X}) = B^s_{\infty\infty}(\mathcal{X})$.** We previously showed that Algorithm 1 effectively competes against any comparator in the broad class of Besov spaces $B^s_{pq}$. In particular, by classical embedding results, when $p = q = \infty$ one has the identification $\mathscr{C}^s(\mathcal{X}) = B^s_{\infty\infty}(\mathcal{X})$ where $\mathscr{C}^s(\mathcal{X})$ is the set of Hölder continuous functions. A function $f \in \mathscr{C}^s(\mathcal{X})$ with $s \in (0, 1]$ if it satisfies the Hölder condition:

$$|f(x) - f(y)| \leqslant L\|x - y\|^s_\infty \quad \text{for all } x, y \in \mathcal{X}, \tag{8}$$

where $L > 0$ is the smallest such constant, denoted $|f|_s$. For $s > 1$, we extend the definition by requiring that all derivatives $D^m f$ exist and satisfy (8) with exponent $s - \lfloor s \rfloor$ for any multi-indices $m \in \mathbb{N}^d$ such that $|m| = \lfloor s \rfloor$.

We now state a corollary of Theorem 1 for Hölder continuous functions, expressed in terms of the Hölder semi-norm $|f|_s$ and sup norm $\|f\|_\infty$.

**Corollary 1** (Regret against Hölder predictors). *Let $T \geqslant 1$ and $s > 0$. Let $\{\phi_{j_0,k}, \psi_{j,k}\}$ be an S-regular wavelet basis with $S > s$. Under the same assumptions as in Theorem 1, Algorithm 1 has regret bounded for any $f \in \mathscr{C}^s(\mathcal{X})$ as*

$$R_T(f) \leqslant CG\|f\|_\infty \sqrt{T} + CG|f|_s \cdot \begin{cases} \sqrt{T} & \text{if } s > \frac{d}{2}, \\ \log_2(T)\sqrt{T} & \text{if } s = \frac{d}{2}, \\ T^{1-\frac{s}{d}} & \text{if } s < \frac{d}{2}. \end{cases}$$

*where $C = C(C_1, C_2, \lambda, \phi, \psi, s)$ is a constant independent of $T$ and $f$, and depend only on $s$, $p$, the wavelet basis, and Assumption 1.*

We prove Corollary 1 in Appendix B. Our results are minimax-optimal for general convex losses, as established in [34, 35], and improve over the guarantees of [29], which are restricted to functions with at most Lipschitz regularity ($s \leqslant 1$). In contrast, our method adapts to any smoothness level $s > 0$. Moreover, Corollary 1 shows that Algorithm 1 adapts simultaneously to both the smoothness $s$ and the Hölder semi-norm $|f|_s$ of any competitor $f \in \mathscr{C}^s$. As in the Besov case, our algorithm automatically trades off between leveraging higher smoothness $s$ and benefiting from smaller $|f|_s$, see (7). This tradeoff will be discussed and exemplified in the next section.

## 4 Adaptive learning in inhomogeneous regularity regimes

In this section, we extend Algorithm 1 to enable local adaptivity, with a particular focus on settings where the target function exhibits spatially inhomogeneous regularity; see Figure 3 for an illustration. Our method is inspired by the localized chaining approach of [29], and we show that it can adapt to local variations in regularity across a broad class of functions in Besov spaces $B^s_{pq}$. This adaptive procedure also yields improved global regret rates over those of Theorem 1 for exp-concave loss functions with optimal guarantees formally established in Theorem 2.

### 4.1 Adaptive Online Wavelet Regression

We begin by describing the partitionning process we use in our strategy, and we further describe the aggregation procedure leading to our adaptive Algorithm 2.

**Partitioning tree.** A common strategy to construct partitions of $\mathcal{X}$ is via hierarchical refinement, with dyadic partitions being a canonical example. Fix $J_0 \in \mathbb{N}^*$. For each $j_0 \in [J_0]$, let $\mathcal{D}_{j_0} = \mathcal{D}_{j_0}(\mathcal{X})$ denote the collection of dyadic subcubes of $\mathcal{X}$ at resolution level $j_0$, where each subcube has side

length $|\mathcal{X}|2^{-j_0}$. We define the full multiscale dyadic collection as $\mathcal{D} = \bigcup_{j_0=0}^{J_0} \mathcal{D}_{j_0}$, spanning scales $j_0 = 0, \dots, J_0$. This collection is naturally aligned with a tree structure $\mathcal{T} = \mathcal{T}(\mathcal{D})$ with node set $\mathcal{N}(\mathcal{T})$. Each node $n \in \mathcal{N}(\mathcal{T})$ is associated with a unique cube $\mathcal{X}_n \in \mathcal{D}$ at some level $\mathrm{l}(n) \in [J_0]$, such that $\mathcal{X}_n \in \mathcal{D}_{\mathrm{l}(n)}$. For any fixed scale $j_0 \in [0, J_0]$, the cubes in $\mathcal{D}_{j_0}$ form a uniform partition of $\mathcal{X}$, and each cube $\mathcal{X}_n \in \mathcal{D}_{j_0}$ with $\mathrm{l}(n) = j_0$ has side length $|\mathcal{X}_n| = |\mathcal{X}|2^{-j_0}$. Furthermore, each node $n$ at level $\mathrm{l}(n) = j_0$ has $2^d$ children corresponding to the dyadic subcubes $\mathcal{X}_{n'} \subset \mathcal{X}_n$ at level $\mathrm{l}(n') = j_0 + 1$. Finally, we refer to any subtree $\mathcal{T}' \subset \mathcal{T}$ that shares the same root and whose leaves or terminal nodes $\mathcal{L}(\mathcal{T}')$ form a (potentially non-uniform) partition of $\mathcal{X}$ as a *pruning* of $\mathcal{T}$; see [29, Def. 2]. The goal is to design an algorithm that effectively tracks the best partition induced by such prunings $\mathcal{T}'$.

**Local adaptation via multi-scale expert aggregation.** Our objective is to identify the optimal starting scale $j_0$ *locally* over $\mathcal{X}$, in order to adapt to the spatial variability in function regularity. Intuitively, allowing finer-scale precision in regions with lower regularity can significantly improve prediction accuracy. To this end, we launch a family of *global* predictors $\hat{f}_{j_0}$ of the form (3), each initialized at a different starting scale $j_0 \in [J_0]$ and sharing a common maximum scale $J = \lceil \frac{S}{d\varepsilon} \log_2 T \rceil$. Following the tree structure $\mathcal{T}$ of depth $J_0$ we associate each node $n \in \mathcal{N}(\mathcal{T})$ to starting scale $j_0 = \mathrm{l}(n)$ and a *local* expert predictor $\hat{f}_{n,\mathbf{a}_n} := \hat{f}_{j_0}|_{\mathcal{X}_n}$ as the restriction of the global predictor $\hat{f}_{j_0}$ to the subregion $\mathcal{X}_n$ and whose scaling coefficients are set to $\boldsymbol{\alpha}_{j_0} = \mathbf{a}_n$ in (3). The scaling coefficients $\mathbf{a}_n$ are supported on a grid $\mathcal{A}$ of precision $T^{-1/2}$. The local predictor is then associated with a restricted scaling index set $\bar{\Lambda}_{j_0,n} \subset \bar{\Lambda}_{j_0}$ and wavelet index set $\Lambda_{j,n} \subset \Lambda_j$ for $j \geqslant j_0$, both supported on $\mathcal{X}_n$. For simplicity, we define the tuples $e = (n, \mathbf{a}_n)$ belonging to some expert set $\mathcal{E} \subset \mathcal{N}(\mathcal{T}) \times \mathcal{A}$ whose cardinal is bounded as $|\mathcal{E}| \leqslant |\mathcal{N}(\mathcal{T})||\mathcal{A}|^\lambda$.

At each time $t \geqslant 1$, we define the set of active experts at round $t$ as $\mathcal{E}_t := \{e = (n, \mathbf{a}_n) \in \mathcal{E} : x_t \in \mathcal{X}_n\}$. The prediction is then formed by aggregating the outputs of active local multi-scale experts in $\mathcal{E}_t$, yielding:

$$\hat{f}_t(x_t) = \sum_{e \in \mathcal{E}_t} w_{e,t} [\hat{f}_{e,t}(x_t)]_B, \quad \text{where} \quad w_{e,t} \in [0,1], \quad \sum_{e \in \mathcal{E}_t} w_{e,t} = 1, \tag{9}$$

and $[\cdot]_B = \max(-B, \min(B, \cdot))$ denotes the clipping operator in $[-B, B]$. Each localized expert $\hat{f}_{e,t} = \hat{f}_{n,\mathbf{a}_n,t}$ is trained independently using Algorithm 1, with scaling coefficients initialized at some $\mathbf{a}_n$, and contributes only within its local region $\mathcal{X}_n$. This framework mirrors an instance of the *sleeping expert* problem, as described for example in [18], and requires a standard sleeping reduction, such as the one in line 4, and then used in lines 5-7 of Algorithm 2. The weights $(w_{e,t})_{e \in \mathcal{E}}$ are updated over time in line 7 using a `weight` procedure based on gradients $\nabla_t \in [-\tilde{G}, \tilde{G}]^{|\mathcal{E}|}$ that satisfies Assumption 2. The overall procedure is summarized in Algorithm 2.

---

**Algorithm 2:** Adaptive Online Wavelet Regression

**Input** : Bounds $G, B > 0$; Set of experts $\mathcal{E}$;
Initial uniform weights $\tilde{\mathbf{w}}_1 = (\tilde{w}_{e,1})_{e \in \mathcal{E}}$; Initial prediction functions $(\hat{f}_{e,1})_{e \in \mathcal{E}}$;

1 **for** $t = 1$ **to** $T$ **do**
2 $\quad$ Receive $x_t$;
3 $\quad$ Reveal active expert set $\mathcal{E}_t$ and local active index set

$$\Gamma_{e,t} := \Gamma_t \cap \bar{\Lambda}_{j_0,n} \cap \cup_{j \geqslant j_0}^J \Lambda_{j,n}, \quad \text{for every } e = (n, \mathbf{a}_n) \in \mathcal{E}_t, \ j_0 = \mathrm{l}(n)$$

$\quad$ with $\Gamma_t$ as in (4);
4 $\quad$ Reduce weights $w_{e,t} \leftarrow \tilde{w}_{e,t} / \sum_{e \in \mathcal{E}_t} \tilde{w}_{e,t}$ if $e \in \mathcal{E}_t$ and $w_{t,e} = 0$ else ;
5 $\quad$ Predict $\hat{f}_t(x_t) = \sum_{e \in \mathcal{E}_t} w_{e,t} [\hat{f}_{e,t}(x_t)]_B$ using active weights $(w_{e,t})_{e \in \mathcal{E}_t}$ ;
6 $\quad$ Reveal gradient $\nabla_t = \nabla_{\tilde{\mathbf{w}}_t} \ell_t \big( \sum_{e \in \mathcal{E}_t} \tilde{w}_{e,t} [\hat{f}_{e,t}(x_t)]_B + \sum_{e \notin \mathcal{E}_t} \tilde{w}_{e,t} \hat{f}_t(x_t) \big)$;
7 $\quad$ Update $\tilde{\mathbf{w}}_{t+1} \leftarrow \texttt{weight}(\tilde{\mathbf{w}}_t, \nabla_t)$ with `weight` satisfying Assumption 2 ;
8 $\quad$ **for** $e \in \mathcal{E}_t$ **do**
9 $\quad\quad$ Reveal gradient $\mathbf{g}_{e,t} = (g_{j,k,t})_{(j,k) \in \Gamma_{e,t}}$ as in (5), on active index set $\Gamma_{e,t}$;
10 $\quad\quad$ Update $\hat{f}_{e,t}$ using Algorithm 1 with input $\Gamma_{e,t}$ and $\mathbf{g}_{e,t}$;

**Output :** $\hat{f}_{T+1} = \sum_{e \in \mathcal{E}} w_{e,T+1} [\hat{f}_{e,T+1}]_B$

---

**Assumption on the aggregation algorithm.** Our method relies on any expert-aggregation algorithm satisfying a second-order regret bound, stated in Assumption 2. State-of-the-art aggregation

algorithms, such as those proposed in [18, 26, 41], satisfy this second-order regret bound and are compatible with the sleeping expert setting.

**Assumption 2.** *Let $\nabla_1, \ldots, \nabla_T \in [-\tilde{G}, \tilde{G}]^{|\mathcal{E}|}$ for $T \geqslant 1$ and $\tilde{G} > 0$. Assume that the weight vectors $\tilde{\mathbf{w}}_t = (\tilde{w}_{e,t})_{e \in \mathcal{E}}$, initialized with a uniform distribution $\tilde{\mathbf{w}}_1$, are updated via the* `weight` *function in Algorithm* 2 *and satisfy the following second-order regret bound:*

$$\sum_{t=1}^{T} \nabla_t^\top \tilde{\mathbf{w}}_t - \nabla_{e,t} \leqslant C_3 \sqrt{\log(|\mathcal{E}|) \sum_{t=1}^{T} (\nabla_t^\top \tilde{\mathbf{w}}_t - \nabla_{e,t})^2} + C_4 \tilde{G},$$

*for all $e \in \mathcal{E}$, where $C_3, C_4 > 0$ are constants.*

Note that for Assumption 2 to hold, the loss gradients $\nabla_t$ must be uniformly bounded in the sup-norm by $\tilde{G}$. Indeed, this is ensured by two factors: first, we consider oracle prediction rules in $L^\infty(\mathcal{X})$, ensuring that their outputs are also uniformly bounded, and second the predictions produced in (9) are clipped to a bounded range.

### 4.2 Regret guarantees under spatially inhomogeneous smoothness (Alg. 2)

**Local Besov regularity.** Let $f \in B_{pq}^s$ for some fixed $1 \leqslant p, q \leqslant \infty$ and $s > \frac{d}{p}$. Let $\mathcal{T}'$ be a pruning of $\mathcal{T}$ and $(\mathcal{X}_n)_{n \in \mathcal{L}(\mathcal{T}')}$ be the associated partition of $\mathcal{X}$. To model spatially varying smoothness,

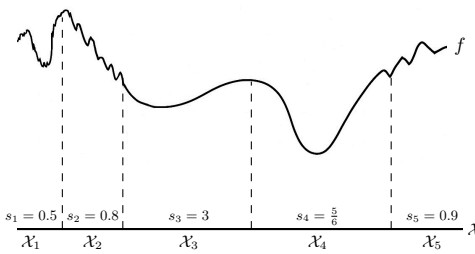

we define the local Besov regularity of a function $f$ over each region $\mathcal{X}_n$ as

$$s_n := \sup \left\{ \alpha : f_{|\mathcal{X}_n} \in B_{pq}^\alpha(\mathcal{X}_n) \right\} \geqslant s, \quad (10)$$

where the restriction $f_{|\mathcal{X}_n}$ belongs to the Besov space $B_{pq}^{s_n}$ over the domain $\mathcal{X}_n$ for fixed global parameters $1 \leqslant p, q \leqslant \infty$. For each region, we denote by $\|f\|_{s_n}$ the corresponding local Besov norm (6). More generally, one could define 'fully' local Besov spaces via triplets $(s_n, p_n, q_n)$, allowing both the smoothness and the integrability parameters to vary across regions. We leave the analysis of such fully adaptive schemes to future work, and we focus on local adaptation in terms of smoothness only.

Figure 3: Example of inhomogeneous function ($d = 1$).

We prove a regret bound for Algorithm 2, expressed in terms of the local regularity of any competitor in a Besov space and that achieves minimax-optimal rates with convex or exp-concave loss functions.

**Theorem 2.** *Let $T \geqslant 1, 1 \leqslant p, q \leqslant \infty, s > \frac{d}{p} > \varepsilon$. Let $f \in B_{pq}^s(\mathcal{X})$ and $B \geqslant \|f\|_\infty$. Let $\mathcal{T}'$ be any pruning of $\mathcal{T}$, together with a collection of local smoothness indices $(s_n)_{n \in \mathcal{L}(\mathcal{T}')}$ and of local norms $(\|f\|_{s_n})_{n \in \mathcal{L}(\mathcal{T}')}$ defined as in (10). Then, under the same assumptions of Theorem 1 and Assumption 2, Algorithm 2 satisfies*

$$R_T(f) \lesssim G \sum_{n \in \mathcal{L}(\mathcal{T}')} \left( B^{1 - \frac{d}{2s_n}} \left( 2^{-1(n)s_n} \|f\|_{s_n} \right)^{\frac{d}{2s_n}} \sqrt{|T_n|} \mathbb{1}_{s_n \geqslant \frac{d}{2}} \right.$$

$$\left. + \left( 2^{-1(n)s_n} \|f\|_{s_n} |T_n|^{1 - \frac{s_n}{d}} \right) \mathbb{1}_{s_n < \frac{d}{2}} + B\sqrt{|T_n|} \right)$$

*and moreover we also have, if $(\ell_t)$ are exp-concave:*

$$R_T(f) \lesssim G \sum_{n \in \mathcal{L}(\mathcal{T}')} \left( B^{1 - \frac{2d}{2s_n + d}} \left( 2^{-1(n)s_n} \|f\|_{s_n} \right)^{\frac{2d}{2s_n + d}} |T_n|^{\frac{d}{2s_n + d}} \mathbb{1}_{s_n \geqslant \frac{d}{2}} \right.$$

$$\left. + 2^{-1(n)s_n} \|f\|_{s_n} |T_n|^{1 - \frac{s_n}{d}} \mathbb{1}_{s_n < \frac{d}{2}} + B \right),$$

*where $\lesssim$ hides logarithmic factors in $T$, and constants independent of $f$ or $T$.*

The proof of Theorem 2 is deferred to Appendix C. Taking $\mathcal{T}'$ as the pruning corresponding to the root of $\mathcal{T}$, Theorem 2 yields minimax-optimal rates in the case $s = \min_n s_n > \frac{d}{p}$, both for convex and exp-concave losses simultaneously. Importantly, the local adaptivity of Algorithm 2 is reflected in the regret bounds in Theorem 2, which now depend on the *local* Besov regularity of the target function. This is especially advantageous in inhomogeneous settings where the function alternates between smooth and highly irregular regions; see Figure 3 for an illustrative example. In such cases, our approach can substantially improve the overall regret compared to classical global adaptive methods that aim to recover the largest (but worst case) smoothness exponent $s$ such that $f \in B_{pq}^s(\mathcal{X})$, assuming the semi-norm $\|f\|_{B_{pq}^s}$ is uniformly bounded.

**Adaptive trade-off between smoothness and norm.** To illustrate the benefit of our strategy, consider the function $f : x \in [0,1] \mapsto x^s$, with $s \in (1,2)$. We have $f \in \mathscr{C}^s(\mathcal{X})$, with global semi-norm $|f|_s < \infty$ (as defined in (8)). However, the semi-norm $|f|_s$ becomes large near $x = 0$ due to the unbounded second derivative, which equals $s(s-1)x^{s-2}$ with $s < 2$ and explodes when $x \to 0$. As a result, this directly affects the regret bound of non-local algorithms. Now consider a partition $\mathcal{X}_1 = [0,\delta]$ and $\mathcal{X}_2 = (\delta,1]$ induced by some pruning, with $\delta = 2^{-j_0}$ for some $j_0 \geqslant 1$. The function $f$ belongs to $\mathscr{C}^s(\mathcal{X}_1)$ near $x = 0$, and to $\mathscr{C}^2(\mathcal{X}_2)$ with bounded semi-norm $|f|_2$ over $\mathcal{X}_2$. Estimating $f$ under this higher regularity on $\mathcal{X}_2$ yields improved guarantees. Our adaptive algorithm automatically exploits this spatial inhomogeneity by focusing on the relevant local smoothness *and* local semi-norm, leading to improved overall performance. Indeed, applying Theorem 2 to functions in $\mathscr{C}^s(\mathcal{X})$, with $s > \frac{1}{2}$ and exp-concave losses, yields:

$$R_T(f) \lesssim (|f|_s|\mathcal{X}_1|^s)^{\frac{2}{2s+1}}|T_1|^{\frac{1}{2s+1}} + (|f|_2|\mathcal{X}_2|^2)^{\frac{2}{5}}|T_2|^{\frac{1}{5}}, \tag{11}$$

where $|f|_2 = \sup_{x \in (\delta,1]}|f''(x)| = s(s-1)\delta^{s-2}$, and $|f|_s < \infty$ by definition. Equation (11) illustrates a trade-off: if $|T_1|$ is negligible — i.e., only a small fraction of the data falls near $0$ — then the second term dominates, and we obtain a regret rate of $O(T^{1/5})$. Conversely, we incur the worst-case rate $O(T^{1/(2s+1)})$, but it is diluted by the small measure of $|\mathcal{X}_1| = \delta$. Finally, consider the case where the data are uniformly distributed, i.e., $|T_1| \approx \delta T$ and $|T_2| \approx (1 - \delta)T$. We obtain:

$$R_T(f) \lesssim \delta^{\frac{2s}{2s+1}}(\delta T)^{\frac{1}{2s+1}} + \delta^{\frac{2(s-2)}{5}}(1-\delta)T^{\frac{1}{5}} \leqslant \delta T^{\frac{1}{2s+1}} + \delta^{\frac{2(s-2)}{5}}T^{\frac{1}{5}}.$$

Optimizing over $\delta$, which is automatically handled by our procedure that selects the best pruning, yields a regret rate of $O(T^r)$ with $r \in ({}^1\!/_5, {}^1\!/_{(2s+1)})$, improving upon the worst-case rate $O(T^{1/(2s+1)})$ achieved by any non-local algorithm.

## Conclusion and perspectives

We proposed adaptive wavelet-based algorithms and analyzed them in the competitive online learning framework against comparator functions in general Besov spaces. Our algorithms achieve minimax-optimal regret guarantees while adapting simultaneously to the regularity of the target function, the convexity properties of the loss functions, and spatially inhomogeneous smoothness - resulting in significant improvements over globally-tuned methods. A limitation of our algorithm, which is common with wavelet-based methods, is the exponential increase in dimensional complexity with the regularity parameter $S$. An interesting parallel can be drawn with traditional wavelet thresholding methods [15] used in the batch setting: in an online manner, our parameter-free algorithm implicitly mimics their behavior by selectively updating coefficients across scales, without requiring explicit thresholds or prior knowledge of the function's regularity. Finally, like most prior work, we focused on functions embedded in $L^\infty$ (i.e., with $s > \frac{d}{p}$). A natural and compelling direction for future research is to extend this analysis to competitors in general $L^p$ spaces with $p < \infty$, which would likely require new analytical tools beyond those used here.

## Acknowledgments

The authors would like to thank Stéphane Jaffard and Albert Cohen for insightful discussions and suggestions on wavelet and approximation theory. P.L. is supported by a PhD scholarship from the Sorbonne Center for Artificial Intelligence (SCAI), Paris.

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

# APPENDIX

## Contents

# A  Proof of Theorem 1

Let $1 \leqslant p, q \leqslant \infty, 0 < s < S$ and $f \in \arg\min_{f \in B_{pq}^s(\mathcal{X})} \sum_{t=1}^T \ell_t(f(x_t))$ the best function to fit the $T$ data over $\mathcal{X} \times [-B, B]$. We start the proof with a decomposition of regret, with any oracle function $f^* \in \mathbb{R}^{\mathcal{X}}$, as

$$
\begin{aligned}
R_T(f) &= \sum_{t=1}^T \ell_t(\hat{f}_t(x_t)) - \ell_t(f(x_t)) \\
&= \sum_{t=1}^T \ell_t(\hat{f}_t(x_t)) - \ell_t(f^*(x_t)) \quad + \quad \sum_{t=1}^T \ell_t(f^*(x_t)) - \ell_t(f(x_t)) \\
&= \text{estimation regret} \quad + \quad \text{approximation regret}.
\end{aligned}
$$

**Nonlinear oracle.** Let $j_0 \in \mathbb{N}$ and $J \geqslant j_0$ to be optimized. We first recall that we use a wavelet development defined for any $J \geqslant j_0$ as

$$
f_J(x) := \sum_{k \in \bar{\Lambda}_{j_0}} \alpha_{j_0, k} \phi_{j_0, k}(x) + \sum_{j = j_0}^J \sum_{k \in \Lambda_j} \beta_{j, k} \psi_{j, k}(x), \tag{12}
$$

with $\alpha_{j_0, k} = \langle f, \phi_{j_0, k} \rangle$ and $\beta_{j, k} = \langle f, \psi_{j, k} \rangle, j \geqslant j_0$, is a truncated wavelet expansion up to level $J \geqslant j_0$. Using a truncated approximation $f_J$ with a large value of $J$ in (12) can lead to suboptimal regret performance, as it requires estimating a large number of wavelet coefficients, thereby incurring a high estimation error. To address this, we introduce a nonlinear oracle $f^*$ that depends only on a selected subset of coefficients across the $J$ levels. This approach, known as best-term or nonlinear approximation, is surveyed in the textbook [12], while constructions close to us in spirit can be found in [15, 16]. We now make this oracle explicit and show that it balances approximation and estimation errors, in particular achieving minimax optimality in our setting. We define the oracle as

$$
f^* = f_{J^*} + f_{\Lambda^*}, \tag{13}
$$

where $f_{J^*}$ is a truncated wavelet expansion up to level $J^* \leqslant J$, as in (12) (i.e. we keep all the detail coefficients up to level $J^*$), and the nonlinear part

$$
f_{\Lambda^*} = \sum_{(j,k) \in \Lambda^*} \beta_{j,k} \psi_{j,k}, \quad \Lambda^* \subset \{(j, k) : k \in \Lambda_j, \ J^* < j \leqslant J\},
$$

uses only wavelet coefficients indexed by an oracle set $\Lambda^*$ drawn from the finer scales $j \in (J^*, J]$. The cardinality of $\Lambda^*$, i.e., the number of retained coefficients, will be optimized in the analysis.

Intuition: The component $f_{\Lambda^*}$ of the oracle consists of the $|\Lambda^*|$ largest coefficients chosen adaptively from the fine scales greater than $J^*$. The procedure is termed *nonlinear* because the choice of coefficients varies with the function $f$, rather than being a fixed linear rule, contrary to the first $J^*$ levels which keep all coefficients independently of the function.

For any $k \in \Lambda_j, j > j_0$, define $v_{j,k} := \beta_{j,k} 2^{js'}$ with $s' = s + \frac{d}{2} - \frac{d}{p}$ as in (6). Observe that the definition of the Besov norm (6) allows a control over the set $\{v_{j,k} : k \in \Lambda_j, J^* < j \leqslant J\}$ in terms of $\ell^p$-norm since

$$
\sum_{J^* < j \leqslant J} \sum_{k \in \Lambda_j} |v_{j,k}|^p \leqslant (J - J^*)^{(1 - \frac{p}{q})_+} \left( \sum_{J^* < j \leqslant J} \left( \sum_{k \in \Lambda_j} |v_{j,k}|^p \right)^{\frac{q}{p}} \right)^{\frac{p}{q}} \leqslant \left[ (J - J^*)^{(\frac{1}{p} - \frac{1}{q})_+} \|f\|_{B_{pq}^s} \right]^p =: C_f^p, \tag{14}
$$

by Hölder's inequality if $q > p$, else by convexity with $q \leqslant p$.

Let $\Lambda^*$ denote the set of indices corresponding to the $|\Lambda^*|$ largest wavelet coefficients (in absolute value) among all $(v_{j,k})$ with $j \in [J^* + 1, J]$ and $k \in \Lambda_j$. The cardinality $|\Lambda^*|$ - that is, the number of wavelet coefficients retained in the nonlinear component of the oracle estimator (13)—will be selected later in the analysis as a tuning parameter. Let $j > J^*$. We have that

$$
|\Lambda^*| \cdot \min_{(j,k) \in \Lambda^*} |v_{j,k}|^p \leqslant \sum_{(j,k) \in \Lambda^*} |v_{j,k}|^p \leqslant C_f^p < \infty,
$$

and in particular since $\forall (j, k) \notin \Lambda^*, |v_{j,k}| \leqslant \min_{(j',k') \in \Lambda^*} |v_{j',k'}|$ one has

$$
\forall (j, k) \notin \Lambda^*, \ |\Lambda^*||v_{j,k}|^p \leqslant C_f^p \implies \forall (j, k) \notin \Lambda^*, \ |\beta_{j,k}| \leqslant C_f 2^{-js'} |\Lambda^*|^{-\frac{1}{p}}. \tag{15}
$$

We are now ready to analyze the regret in two steps—that is an *estimation error* and an *approximation error*.

**Step 1: Bounding the estimation regret.** We set

$$R_1 := \sum_{t=1}^{T} \ell_t(\hat{f}_t(x_t)) - \ell_t(f^*(x_t)) = \sum_{t=1}^{T} \ell_t\left(\sum_{(j,k)} c_{j,k,t}\varphi_{j,k}(x_t)\right) - \ell_t\left(\sum_{(j,k)} c_{j,k}\varphi_{j,k}(x_t)\right),$$

where the sum is over all scaling and detail coefficients with indices in $\{(j_0,k): k \in \bar{\Lambda}_{j_0}\} \cup \{(j,k): j \geqslant j_0, \ k \in \Lambda_j\}$, where $c_{j,k}$ stands for either the scaling coefficient $\alpha_{j_0,k}$ or the detail coefficient $\beta_{j,k}$ (and their sequential counterparts $c_{j,k,t}$ depend on $t$), and $\varphi_{j,k}$ denotes either the scaling function $\phi_{j_0,k}$ or the wavelet function $\psi_{j,k}$.

Since $\hat{y} \mapsto \ell_t(\hat{y})$ is convex and both $\hat{f}_t, f^*$ are linear in the $\{c_{j,k}\}$, then $\ell_t \circ \hat{f}$ and $\ell_t \circ f^*$ are convex in $\{c_{j,k}\}$ and we have by convexity:

$$R_1 \leqslant \sum_{t=1}^{T} \sum_{j,k} g_{j,k,t}(c_{j,k,t} - c_{j,k}),$$

where $g_{j,k,t} = \ell_t'(\hat{f}_t(x_t))\varphi_{j,k}(x_t)$ by Equation (5). Observe that $\max_t g_{j,k,t} \leqslant 2^{\frac{dj}{2}} G\|\varphi\|_\infty =: \hat{G}_j$ for any $j, k$. Then, first by Assumption 1, and second by the structure of the oracle (13) — namely, that $\forall j > J^*$ such that $(j,k) \notin \Lambda^*$, we have $c_{j,k} = 0$ — we get:

$$R_1 \leqslant \sum_{j,k} \sum_{t=1}^{T} g_{j,k,t}(c_{j,k,t} - c_{j,k})$$

$$\leqslant \sum_{j,k} |c_{j,k}| \left(C_1\sqrt{\sum_{t=1}^{T}|g_{j,k,t}|^2} + C_2\hat{G}_j\right)$$

$$= \underbrace{\sum_{j\leqslant J^*, k} |c_{j,k}| \left(C_1\sqrt{\sum_{t=1}^{T}|g_{j,k,t}|^2} + C_2\hat{G}_j\right)}_{:=R_1(f_{J^*})} + \underbrace{\sum_{\substack{(j,k)\in\Lambda^* \\ j>J^*}} |c_{j,k}| \left(C_1\sqrt{\sum_{t=1}^{T}|g_{j,k,t}|^2} + C_2\hat{G}_j\right)}_{:=R_1(f_{\Lambda^*})} \quad (16)$$

where $C_1, C_2 > 0$ are factors (possibly including $\log T$; see Assumption 1). The estimation regret $R_1$ is thus controlled in (16) by a sum of individual regrets over the nonzero coefficients $c_{j,k}$ that define $f^* = f_{J^*} + f_{\Lambda^*}$. The sum naturally splits into two parts: the linear part $R_1(f_{J^*})$ over $\{(j_0,k): k \in \bar{\Lambda}_{j_0}\} \cup \{(j,k): j_0 \leqslant j \leqslant J^*, \ k \in \Lambda_j\}$, and the nonlinear part $R_1(f_{\Lambda^*})$ over the indices in $\Lambda^*$.

- **Linear part: bounding $R_1(f_{J^*})$.** The wavelet basis $\{\phi_{j_0,k}, \psi_{j,k}\}$ is assumed to be $S$-regular with $S > s$, so we can invoke the characterization of Besov spaces with $\|f\|_{B_{pq}^s} < \infty$ (see Eq. (6)). Let $p, p' \geqslant 1$ be such that $\frac{1}{p} + \frac{1}{p'} = 1$. Applying Hölder's inequality to the detail coefficients at levels $j \in [j_0, J^*]$, we obtain, with $\hat{G}_j = G2^{\frac{jd}{2}}\|\psi\|_\infty, j \geqslant j_0$:

$$\sum_{j_0\leqslant j\leqslant J^*} \sum_{k\in\Lambda_j} |\beta_{j,k}|\left(C_1\sqrt{\sum_{t=1}^{T}|g_{j,k,t}|^2} + C_2G\|\psi\|_\infty 2^{\frac{jd}{2}}\right)$$

$$\leqslant \sum_{j_0\leqslant j\leqslant J^*} \left(\sum_{k\in\Lambda_j}|\beta_{j,k}|^p\right)^{\frac{1}{p}}\left(C_1\left(\sum_{k\in\Lambda_j}\left(\sqrt{\sum_{t=1}^{T}|g_{j,k,t}|^2}\right)^{p'}\right)^{\frac{1}{p'}} + C_2G\|\psi\|_\infty 2^{\frac{jd}{2}}|\Lambda_j|^{\frac{1}{p'}}\right)$$

$$= \sum_{j_0\leqslant j\leqslant J^*} \|\boldsymbol{\beta}_j\|_p\left(C_1\sqrt{\left(\sum_{k\in\Lambda_j}\left(\sum_{t=1}^{T}|g_{j,k,t}|^2\right)^{\frac{p'}{2}}\right)^{\frac{2}{p'}}} + C_2G\|\psi\|_\infty 2^{\frac{jd}{2}}|\Lambda_j|^{\frac{1}{p'}}\right)$$

$$\leqslant \sum_{j_0\leqslant j\leqslant J^*} \|\boldsymbol{\beta}_j\|_p\left(C_1|\Lambda_j|^{(\frac{1}{2}-\frac{1}{p})_+}\sqrt{\sum_{k\in\Lambda_j}\sum_{t=1}^{T}|g_{j,k,t}|^2} + C_2G2^{\frac{jd}{2}}|\Lambda_j|^{1-\frac{1}{p}}\right)$$

$$(17)$$

where the last inequality uses $\|x\|_{\frac{p'}{2}} \leqslant |\Lambda_j|^{(\frac{2}{p'}-1)_+}\|x\|_1$ for a vector $x$ of dimension $|\Lambda_j|$ and $(\cdot)_+ := \max\{\cdot, 0\}$.

Repeating for the scaling coefficients for $k \in \bar{\Lambda}_{j_0}$, summing in $j = j_0, \ldots, J^*$ and bounding $|\Lambda_j| \leqslant \lambda 2^{dj}$ and $|\bar{\Lambda}_{j_0}| \leqslant \lambda 2^{dj_0}$, we get:

$$R_1(f_{J^*}) \leqslant \|\boldsymbol{\alpha}_{j_0}\|_p\left(C_1\lambda 2^{dj_0(\frac{1}{2}-\frac{1}{p})_+}\sqrt{\sum_{k\in\Lambda_j}\sum_{t=1}^{T}|g_{j_0,k,t}|^2} + C_2G\|\phi\|_\infty 2^{\frac{j_0d}{2}}\lambda 2^{dj_0(1-\frac{1}{p})}\right)$$

$$+ \sum_{j=j_0}^{J^*} \|\boldsymbol{\beta}_j\|_p\left(C_1\lambda 2^{dj(\frac{1}{2}-\frac{1}{p})_+}\sqrt{\sum_{k\in\Lambda_j}\sum_{t=1}^{T}|g_{j,k,t}|^2} + C_2G\|\psi\|_\infty 2^{\frac{jd}{2}}\lambda 2^{dj(1-\frac{1}{p})}\right) \quad (18)$$

where we recall the scaling coefficients are $\boldsymbol{\alpha}_{j_0} = (\alpha_{j_0,k})$ and the detail coefficients at scale $j$ are $\boldsymbol{\beta}_j = (\beta_{j,k})$.

On the other hand, over each level $j \geqslant j_0$, one has

$$\sqrt{\sum_{k \in \Lambda_j} \sum_{t=1}^{T} |g_{j,k,t}|^2} = \sqrt{\sum_{k \in \Lambda_j} \sum_{t=1}^{T} |\ell_t'(\hat{f}_t(x_t)) \psi_{j,k}(x_t)|^2}$$

$$\leqslant G \sqrt{\sum_{k \in \Lambda_j} \sum_{t=1}^{T} |\psi_{j,k}(x_t)|^2}$$

$$= G 2^{\frac{dj}{2}} \sqrt{\sum_{t=1}^{T} \sum_{k \in \Lambda_j} |\psi(2^j x_t - k)|^2}, \tag{19}$$

where we used the fact that $|\ell'(\hat{f}_t(x_t))| \leqslant G$ (since $\hat{y} \mapsto \ell_t(\hat{y})$ is $G$-Lipschitz), the definition of $\psi_{j,k}$ and we applied Jensen's inequality. Equation (19) also holds for the scaling level, replacing $\psi_{j,k}$ by $\phi_{j_0,k}$ over the index set $\bar{\Lambda}_{j_0}$.

By D.2, one has

$$\sup_x \sum_k |\phi(x-k)|^2 \leqslant M_\phi \|\phi\|_\infty \quad \text{and} \quad \sup_x \sum_k |\psi(x-k)|^2 \leqslant M_\psi \|\psi\|_\infty.$$

With $1 - \frac{1}{p} \leqslant \frac{1}{2} + (\frac{1}{2} - \frac{1}{p})_+$ we get from (18) and (19)

$$R_1(f_{J^*}) \leqslant \lambda G \left[ \left( C_1 (M_\phi \|\phi\|_\infty)^{\frac{1}{2}} \sqrt{T} + C_2 \|\phi\|_\infty 2^{\frac{d}{2} j_0} \right) \|\boldsymbol{\alpha}_{j_0}\|_p 2^{dj_0(\frac{1}{2} + (\frac{1}{2} - \frac{1}{p})_+)} \right.$$

$$\left. \left( C_1 (M_\psi \|\psi\|_\infty)^{\frac{1}{2}} \sqrt{T} + C_2 \|\psi\|_\infty 2^{\frac{d}{2} J^*} \right) \sum_{j=j_0}^{J^*} \|\boldsymbol{\beta}_j\|_p 2^{dj(\frac{1}{2} + (\frac{1}{2} - \frac{1}{p})_+)} \right]. \tag{20}$$

Then since $\|f\|_{B_{pq}^s} < \infty$ in (6), we apply Hölder's inequality with $q, q' \geqslant 1$ that entails

$$\sum_{j=j_0}^{J^*} \|\boldsymbol{\beta}_j\|_p 2^{jd(\frac{1}{2} + (\frac{1}{2} - \frac{1}{p})_+)} = \sum_{j=j_0}^{J} 2^{-j(s + \frac{d}{2} - \frac{d}{p})} 2^{j(s + \frac{d}{2} - \frac{d}{p})} \|\boldsymbol{\beta}_j\|_p 2^{jd(\frac{1}{2} + (\frac{1}{2} - \frac{1}{p})_+)}$$

$$\leqslant \left( \sum_{j=j_0}^{J^*} 2^{-jq'(s - \frac{d}{p} - d(\frac{1}{2} - \frac{1}{p})_+)} \right)^{\frac{1}{q'}} \left( \sum_{j=j_0}^{J} 2^{jq(s + \frac{d}{2} - \frac{d}{p})} \|\boldsymbol{\beta}_j\|_p^q \right)^{\frac{1}{q}}$$

$$\leqslant \|f\|_{B_{pq}^s} \sum_{j=0}^{J^*} 2^{-j(s - \frac{d}{p} - d(\frac{1}{2} - \frac{1}{p})_+)}, \quad \text{since } \|\cdot\|_{q'} \leqslant \|\cdot\|_1, q' \geqslant 1.$$

Finally, we get from (20) with $\|\boldsymbol{\alpha}_{j_0}\|_p \leqslant \|f\|_{B_{pq}^s}$:

$$R_1(f_{J^*}) \leqslant \lambda G \|f\|_{B_{pq}^s} M \left( \left( C_1 \sqrt{T} + C_2 2^{\frac{d}{2} j_0} \right) 2^{dj_0(\frac{1}{2} + (\frac{1}{2} - \frac{1}{p})_+)} \right.$$

$$\left. + \left( C_1 \sqrt{T} + C_2 2^{\frac{d}{2} J^*} \right) \sum_{j=j_0}^{J^*} 2^{-j\beta} \right), \tag{21}$$

with $\beta := s - \frac{d}{p} - d(\frac{1}{2} - \frac{1}{p})_+$ and $M := \left( \max(M_\phi \|\phi\|_\infty, M_\psi \|\psi\|_\infty, \|\phi\|_\infty^2, \|\psi\|_\infty^2) \right)^{\frac{1}{2}} < \infty$

- **Nonlinear part: bounding $R_1(f_{\Lambda^*})$.** Let $\Lambda^* = \cup_{j=J^*+1}^{J} \Lambda_j^*$ where $|\Lambda_j^*| \leqslant |\Lambda_j|$ is now the oracle sparse set made of positions $k$ at level $j$. One has by (17) on the levels $j = J^* + 1, \ldots, J$,

$$R_1(f_{\Lambda^*}) \leqslant \sum_{J^* < j \leqslant J} \|\boldsymbol{\beta}_j\|_p \left( C_1 |\Lambda_j^*|^{(\frac{1}{2} - \frac{1}{p})_+} \sqrt{\sum_{k \in \Lambda_j^*} \sum_{t=1}^{T} |g_{j,k,t}|^2} + C_2 G 2^{\frac{jd}{2}} \|\psi\|_\infty |\Lambda_j^*|^{1 - \frac{1}{p}} \right)$$

$$\leqslant G M \sum_{J^* < j \leqslant J} \|\boldsymbol{\beta}_j\|_p \left( C_1 |\Lambda_j^*|^{(\frac{1}{2} - \frac{1}{p})_+} 2^{\frac{dj}{2}} \sqrt{T} + C_2 2^{\frac{jd}{2}} |\Lambda_j^*|^{1 - \frac{1}{p}} \right)$$

where second inequality follows from (19). Then, using Hölder's inequality with $q \geqslant 1$ one has

$$\sum_{J^* < j \leqslant J} 2^{\frac{dj}{2}} \|\boldsymbol{\beta}_j\|_p |\Lambda_j^*|^{(\frac{1}{2} - \frac{1}{p})_+} = \sum_{J^* < j \leqslant J} 2^{j(s + \frac{d}{2} - \frac{d}{p})} \|\boldsymbol{\beta}_j\|_p \cdot 2^{-j(s - \frac{d}{p})} |\Lambda_j^*|^{(\frac{1}{2} - \frac{1}{p})_+}$$

$$\leqslant \|f\|_{B_{pq}^s} \sum_{J^* < j \leqslant J} 2^{-j(s - \frac{d}{p})} |\Lambda_j^*|^{(\frac{1}{2} - \frac{1}{p})_+}.$$

Finally, since $\sum_{J^* < j \leqslant J} |\Lambda_j^*| = |\Lambda^*|$, one has

$$\sum_{J^* < j \leqslant J} 2^{-j(s - \frac{d}{p})} |\Lambda_j^*|^{(\frac{1}{2} - \frac{1}{p})_+} \leqslant |\Lambda^*|^{(\frac{1}{2} - \frac{1}{p})_+} \sum_{J^* < j \leqslant J} 2^{-j(s - \frac{d}{p})} \leqslant |\Lambda^*|^{(\frac{1}{2} - \frac{1}{p})_+} \frac{2^{-J^*(s - \frac{d}{p})}}{2^{s - \frac{d}{p}} - 1},$$

by Hölder's inequality in the case $p \geqslant 2$ and since $s > \frac{d}{p}$. Similarly, for the second term we have

$$\sum_{J^* < j \leqslant J} 2^{\frac{dj}{2}} \|\boldsymbol{\beta}_j\|_p |\Lambda_j^*|^{1 - \frac{1}{p}} \leqslant \|f\|_{B_{pq}^s} \frac{2^{-J^*(s - \frac{d}{p})}}{2^{s - \frac{d}{p}} - 1} |\Lambda^*|^{1 - \frac{1}{p}}.$$

All in one, with $|\Lambda^*|^{1 - \frac{1}{p}} \leqslant |\Lambda^*|^{\frac{1}{2} + (\frac{1}{2} - \frac{1}{p})_+}$ one has

$$R_1(f_{\Lambda^*}) \leqslant GM \|f\|_{B_{pq}^s} \frac{C_1 \sqrt{T} + C_2 |\Lambda^*|^{\frac{1}{2}}}{2^{s - \frac{d}{p}} - 1} \cdot 2^{-J^*(s - \frac{d}{p})} |\Lambda^*|^{(\frac{1}{2} - \frac{1}{p})_+}. \tag{22}$$

- Bound on $R_1$: We use (21) and (22) and we reach

$$R_1 \leqslant R_1(f_{J^*}) + R_1(f_{\Lambda^*})$$

$$\leqslant G \|f\|_{B_{pq}^s} M \Big[ \lambda \big( C_1 \sqrt{T} + C_2 2^{\frac{d}{2} j_0} \big) 2^{d j_0 (\frac{1}{2} + (\frac{1}{2} - \frac{1}{p})_+)}$$

$$+ \lambda \big( C_1 \sqrt{T} + C_2 2^{\frac{d}{2} J^*} \big) \sum_{j = j_0}^{J^*} 2^{-j\beta}$$

$$+ \big( C_1 \sqrt{T} + C_2 |\Lambda^*|^{\frac{1}{2}} \big) |\Lambda^*|^{(\frac{1}{2} - \frac{1}{p})_+} \frac{2^{-J^*(s - \frac{d}{p})}}{2^{s - \frac{d}{p}} - 1} \Big], \tag{23}$$

where we recall $\beta := s - \frac{d}{p} - d(\frac{1}{2} - \frac{1}{p})_+$ and $s' = s + \frac{d}{2} - \frac{d}{p}$ and $|\Lambda^*|$ is the number of 'non-linear' coefficients we keep below level $J^*$.

**Step 2: Bounding the approximation regret.** We now bound the term incurred by approximating $f$ by its nonlinear wavelet approximation $f^*$. Using the $G$-Lipschitz property of each loss $\ell_t$ and the uniform bound on the approximation error, we obtain:

$$R_2 := \sum_{t=1}^{T} \big( \ell_t(f^*(x_t)) - \ell_t(f(x_t)) \big) \leqslant G \sum_{t=1}^{T} |f^*(x_t) - f(x_t)| \leqslant GT \|f^* - f\|_\infty. \tag{24}$$

With $f^* = f_{J^*} + f_{\Lambda^*}$ and $f_J$ the truncated wavelet expansion (12) at level $J \geqslant J^* \geqslant j_0$ we have with the triangle inequality

$$R_2 \leqslant GT(\|(f_{J^*} + f_{\Lambda^*}) - f_J\|_\infty + \|f_J - f\|_\infty)$$

First, since $f_{J^*}, f_J$ are both wavelet expansion truncated respectively at level $J^*$ and $J$, one has

$$\|(f_{J^*} + f_{\Lambda^*}) - f_J\|_\infty = \Big\| \sum_{(j,k) \notin \Lambda^*} \beta_{j,k} \psi_{j,k} \Big\|_\infty$$

$$\leqslant \sum_{(j,k) \notin \Lambda^*} \|\beta_{j,k} \psi_{j,k}\|_\infty$$

$$\leqslant \sum_{j = J^* + 1}^{J} 2^{j \frac{d}{2}} \sup_{k:(j,k) \notin \Lambda^*} |\beta_{j,k}| \cdot \| \sum_{k \in \Lambda_j} |\psi(2^j \cdot - k)| \|_\infty \quad \leftarrow \text{ by definition of } \psi_{j,k}$$

$$\leqslant M_\psi C_f |\Lambda^*|^{-\frac{1}{p}} \sum_{j = J^* + 1}^{J} 2^{j \frac{d}{2}} 2^{-j s'} \quad \leftarrow \text{ by Definition D.2 and (15)}$$

$$\leqslant M_\psi C_f |\Lambda^*|^{-\frac{1}{p}} \frac{2^{-J^*(s - \frac{d}{p})}}{2^{s - \frac{d}{p}} - 1} \quad \leftarrow \text{ replacing } s' \text{ and with } s > \frac{d}{p}. \tag{25}$$

Second, with $s > \frac{d}{p}$, using the characterizations of Besov spaces and classical results on Sobolev embeddings (see, e.g., [20, Prop. 4.3.8] or [8, 13]), $B_{pq}^s(\mathcal{X}) \subset B_{\infty\infty}^{s-\frac{d}{p}}(\mathcal{X})$ and one has

$$\|f_J - f\|_\infty \leqslant M_\psi \|f\|_{B_{pq}^s} \sum_{j>J} 2^{-j(s-\frac{d}{p})} \leqslant M_\psi \|f\|_{B_{pq}^s} \frac{2^{-J(s-\frac{d}{p})}}{2^{s-\frac{d}{p}} - 1}, \tag{26}$$

where $s > \frac{d}{p}$.
Finally, with (25) and (26) one has

$$R_2 \leqslant \frac{G M_\psi \|f\|_{B_{pq}^s}}{2^{s-\frac{d}{p}} - 1} \left( 2^{-J(s-\frac{d}{p})} T + (J - J^*)^{(\frac{1}{p}-\frac{1}{q})} + 2^{-J^*(s-\frac{d}{p})} |\Lambda^*|^{-\frac{1}{p}} T \right). \tag{27}$$

**Step 3: Optimization on $J^*, J, |\Lambda^*|$ and conclusion.** Let $j_0 = 0$. From (23) and (27) we reach the following regret bound

$$R_T(f) = R_1 + R_2 \leqslant CG\|f\|_{B_{pq}^s} \Bigg[ \left( C_1\sqrt{T} + C_2 2^{\frac{d}{2}J^*} \right) \left( 1 + \sum_{j=0}^{J^*} 2^{-j\beta} \right)$$

$$+ \left( C_1\sqrt{T} + C_2|\Lambda^*|^{\frac{1}{2}} \right) 2^{-J^*(s-\frac{d}{p})} |\Lambda^*|^{(\frac{1}{2}-\frac{1}{p})+}$$

$$+ 2^{-J(s-\frac{d}{p})} T + (J - J^*)^{(\frac{1}{p}-\frac{1}{q})} + 2^{-J^*(s-\frac{d}{p})} |\Lambda^*|^{-\frac{1}{p}} T \Bigg], \quad (28)$$

with $C$ some constant that can change from a line to another (depending on $\lambda, \|\psi\|_\infty, M, M_\psi, ...$), $\beta = s - \frac{d}{p} - d(\frac{1}{2} - \frac{1}{p})_+$. We keep the explicit dependence on $J, J^*$, and $|\Lambda^*|$, as we now aim to optimize the upper bound with respect to these parameters. We have three different regimes depending on the sign of $\beta$ in (28). Observe that

$$\beta = \begin{cases} s - \frac{d}{2} & \text{if } p \geqslant 2, \\ s - \frac{d}{p} & \text{if } p < 2, \end{cases}$$

and we also have $s > \frac{d}{p}$.

**Case 1: $\beta > 0$.** This regime corresponds to sufficiently regular functions: since $s > \frac{d}{p}$, this corresponds to the case

$$p < 2, \qquad \text{or} \qquad s > \frac{d}{2}.$$

In this case, the geometric sum is bounded by

$$\sum_{j=0}^{J^*} 2^{-j\beta} \leqslant \frac{1}{1 - 2^{-\beta}}.$$

Choosing

$$\begin{cases} J^* = \left\lceil \frac{1}{d} \log_2(T) \right\rceil, \\ |\Lambda^*| = 2^{J^* d}, \\ J = \left\lceil \frac{S}{d\varepsilon} \log_2(T) \right\rceil, \end{cases} \implies \begin{cases} 2^{-J^*(s-\frac{d}{p})} |\Lambda^*|^{-\frac{1}{p}} T = 2^{-sJ^*} T = T^{1-\frac{s}{d}}, \\ 2^{-J^*(s-\frac{d}{p})} |\Lambda^*|^{(\frac{1}{2}-\frac{1}{p})+} \leqslant T^{\frac{1}{2}-\frac{s}{d}}, \\ 2^{-J(s-\frac{d}{p})} T \leqslant T^{1-\frac{s}{d}}, \\ |\Lambda^*|^{\frac{1}{2}} = 2^{\frac{d}{2}J^*} = \sqrt{T} \end{cases} \tag{29}$$

with $s - \frac{d}{p} > \varepsilon > 0$ and $S > s$, and this entails a total regret

$$R_T(f) \leqslant CG\|f\|_{B_{pq}^s} \Bigg[ (C_1 + C_2)\sqrt{T} \left( 2 + \frac{1}{1 - 2^{-\beta}} + T^{\frac{1}{2}-\frac{s}{d}} \right)$$

$$+ T^{1-\frac{s}{d}} \left( 1 + \left( \frac{1}{d} \left( \frac{S}{\varepsilon} - 1 \right) \log_2(T) \right)^{(\frac{1}{p}-\frac{1}{q})+} \right) \Bigg]. \quad (30)$$

With $s \geqslant d/2$, we have $T^{1-s/d} \leqslant \sqrt{T}$ and $R_T(f) = O(G\|f\|_{B_{pq}^s} \sqrt{T})$.
*Remark.* The notation $O(\cdot)$ here hides $\log_2(T)$ factors that appear when $p < q$. This originates from the nonlinear oracle construction in the analysis (see Inequality (14)). In addition, log terms may also be absorbed into the constants $C_1, C_2$ coming from the parameter-free subroutine (see Assumption 1). This remark also holds for the remaining cases.

**Case 2:** $\beta = 0$. This critical regime occurs when $p \geqslant 2$ and $s = \frac{d}{2}$. The sum becomes:

$$\sum_{j=0}^{J^*} 2^{-j\beta} = J^* + 1.$$

Choosing $J^*, |\Lambda^*|$ and $J$ as in (29) yields the bound:

$$R_T(f) \leqslant CG\|f\|_{B_{pq}^s} \left[ (C_1 + C_2)\sqrt{T}\left( 2 + \frac{1}{d}\log_2 T + T^{\frac{1}{2}-\frac{s}{d}} \right) \right.$$
$$\left. + \sqrt{T}\left( 1 + \left( \frac{1}{d}\left( \frac{S}{\varepsilon} - 1 \right)\log_2(T) \right)^{(\frac{1}{p}-\frac{1}{q})+} \right) \right]. \quad (31)$$

That is $R_T(f) = O(G\|f\|_{B_{pq}^s} \log_2(T)\sqrt{T})$.

**Case 3:** $\beta < 0$. This corresponds to the low regularity case: $\beta = s - \frac{d}{2}$ and

$$p \geqslant 2 \qquad \text{and} \qquad \frac{d}{p} < s < \frac{d}{2}.$$

Here, the geometric sum is bounded as:

$$\sum_{j=0}^{J^*} 2^{-j\beta} \leqslant \frac{2^{-J^*\beta}}{2^{-\beta} - 1}.$$

With $J^*, |\Lambda^*|$ and $J$ as in (29), the regret bound becomes:

$$R_T(f) \leqslant CG\|f\|_{B_{pq}^s} \left[ (C_1 + C_2)\sqrt{T}\left( 1 + \frac{T^{-\frac{\beta}{d}}}{2^{-\beta} - 1} + T^{\frac{1}{2}-\frac{s}{d}} \right) \right.$$
$$\left. + T^{1-\frac{s}{d}}\left( 1 + \left( \frac{1}{d}\left( \frac{S}{\varepsilon} - 1 \right)\log_2(T) \right)^{(\frac{1}{p}-\frac{1}{q})+} \right) \right]. \quad (32)$$

With $\sqrt{T}T^{-\frac{\beta}{d}} = \sqrt{T}T^{\frac{1}{2}-\frac{s}{d}} = T^{1-\frac{s}{d}}$, one has $R_T(f) = O(G\|f\|_{B_{pq}^s}T^{1-\frac{s}{d}})$.

# B Proof of Corollary 1

The proof is based on that of Theorem 1, in Appendix A, case $p = q = \infty$.

Let $s > 0, f \in \arg\min_{f \in \mathscr{C}^s(\mathcal{X})} \sum_{t=1}^T \ell_t(f(x_t))$ the best function to fit the $T$ data over $\mathcal{X} \times \mathbb{R}$ and $f^* = f_J$ defined as in (12). One key point is that in the case of Hölder-smooth function ($p = q = \infty$), the nonlinear set $\Lambda^*$ of wavelet coefficients will not be needed to achieve optimal rates.

We start with a decomposition of regret with the oracle $f^* = f_J$ as in the proof of Theorem 1 in Appendix A and we have:

$$R_T(f) = \underbrace{\sum_{t=1}^T \ell_t(\hat{f}_t(x_t)) - \ell_t(f_J(x_t))}_{=:R_1} + \underbrace{\sum_{t=1}^T \ell_t(f_J(x_t)) - \ell_t(f(x_t))}_{=:R_2}$$

**Step 1: Bounding estimation regret $R_1$.** From (16), one has

$$R_1 \leqslant \sum_{k \in \bar{\Lambda}_{j_0}} |\alpha_{j_0,k}| \left( C_1 \sqrt{\sum_{t=1}^T |g_{j,k,t}|^2} + C_2 G \|\phi\|_\infty 2^{\frac{j_0 d}{2}} \right)$$

$$+ \sum_{j=j_0}^J \sum_{k \in \Lambda_j} |\beta_{j,k}| \left( C_1 \sqrt{\sum_{t=1}^T |g_{j,k,t}|^2} + C_2 G \|\psi\|_\infty 2^{\frac{jd}{2}} \right) \quad (33)$$

where $C_1, C_2 > 0$ are relative to Assumption 1, $\alpha_{j,k}$ refers to the scaling coefficients and $\beta_{j,k}$ the detail coefficients.

Since the wavelet basis $\{\phi_{j_0,k}, \psi_{j,k}\}$ is assumed to be $S$-regular with $S > s$ (Definition 2) and $f \in \mathscr{C}^s(\mathcal{X})$, by Proposition 1, the detail coefficients at every level $j \geqslant j_0$ are bounded as:

$$|\beta_{j,k}| = |\langle f, \psi_{j,k} \rangle| \leqslant C(\psi,s) |f|_s 2^{-j(s+d/2)},$$

where $C(\psi, s)$ is a positive constant that only depends on the $S$-regular wavelet basis and $|f|_s$ refers to the semi-norm of $f$ defined in (8).

For the scaling level $j_0$, then for every $k$, one has:

$$|\alpha_{j_0,k}| = |\langle f, \phi_{j_0,k} \rangle| \leqslant \|f\|_\infty \cdot \|\phi_{j_0,k}\|_1 \leqslant 2^{-\frac{j_0 d}{2}} \|\phi\|_1 \|f\|_\infty,$$

where we used

$$\|\phi_{j_0,k}\|_1 \leqslant \int_{\mathbb{R}^d} 2^{j_0 d/2} \phi(2^{j_0} x - k) \, dx \overset{u = 2^{j_0} x - k}{=} 2^{j_0 \frac{d}{2}} 2^{-j_0 d} \int_{\mathbb{R}^d} |\phi(u)| \, du = 2^{-\frac{j_0 d}{2}} \|\phi\|_1$$

and $\|\phi\|_1 < \infty$ since the scaling function $\phi$ is assumed to be localized (e.g. compactly supported).

Then, plugging the above upper bound, we get with $g_{j_0,k,t} \leqslant G \|\phi\|_\infty 2^{\frac{j_0 d}{2}}$ :

$$R_1 \leqslant G \|\phi\|_1 \|f\|_\infty 2^{-\frac{j_0 d}{2}} \cdot |\bar{\Lambda}_{j_0}| \cdot \|\phi\|_\infty \left( C_1 2^{\frac{j_0 d}{2}} \sqrt{T} + C_2 2^{\frac{j_0 d}{2}} \right)$$

$$+ C(\psi,s) |f|_s \sum_{j=j_0}^J 2^{-j(s+d/2)} \left( C_1 \sum_{k \in \Lambda_j} \sqrt{\sum_{t=1}^T |g_{j,k,t}|^2} + C_2 G \|\psi\|_\infty 2^{\frac{jd}{2}} |\Lambda_j| \right),$$

Using Cauchy-Schwarz's inequality as long as the form of the gradients in (5) and the bound (19), we have over each level $j \in [j_0, J]$,

$$\sum_{k \in \Lambda_j} \sqrt{\sum_{t=1}^T |g_{j,k,t}|^2} \leqslant G \lambda^{\frac{1}{2}} 2^{dj} \sqrt{\sum_{t=1}^T \sum_{k \in \Lambda_j} |\psi(2^j x_t - k)|^2}$$

$$\leqslant G (\lambda \|\psi\|_\infty M_\psi)^{\frac{1}{2}} 2^{dj} \sqrt{T}$$

where $\| \sum_{k \in \Lambda_j} |\psi(\cdot - k)|^2 \|_\infty \leqslant \|\psi\|_\infty M_\psi < \infty$ (see D.2) and $|\Lambda_j| = \lambda 2^{dj}$. Finally, with $M = \max(M_\psi \|\psi\|_\infty, \|\phi\|_\infty^2, \|\psi\|_\infty^2)^{\frac{1}{2}}$

$$R_1 \leqslant G M \lambda \left( \|f\|_\infty \|\phi\|_1 2^{j_0 d} (C_1 \sqrt{T} + C_2) + C(\psi,s) |f|_s (C_1 \sqrt{T} + C_2 2^{\frac{d}{2} J}) \sum_{j=j_0}^J 2^{-j(s-d/2)} \right) \quad (34)$$

Setting $j_0 = 0$, the sum can be upper-bounded with 3 different cases as

$$\sum_{j=0}^J 2^{-j(s-\frac{d}{2})} \leqslant \begin{cases} (1 - 2^{-(s-\frac{d}{2})})^{-1} & \text{if } d < 2s, \\ J + 1 & \text{if } d = 2s, \\ 2^{-(J+1)(s-\frac{d}{2})} (2^{-(s-\frac{d}{2})} - 1)^{-1} & \text{if } d > 2s. \end{cases}$$

**Step 2: Bounding the approximation regret.** Following (27), one has:

$$R_2 := \sum_{t=1}^{T} \ell_t(\hat{f}^*(x_t)) - \ell_t(f(x_t)) \leqslant G \sum_{t=1}^{T} |\hat{f}^*(x_t) - f(x_t)| \leqslant GT\|K_J f - f\|_\infty \leqslant C_4 GT|f|_s 2^{-sJ}, \quad (35)$$

where $C_4 = C_4(\psi, s)$ - see [20, Prop. 4.1.5] for instance with assumption D.3.

**Step 3: upper-bounding $R_T(f)$.** We need to balance (34) and (35), and finding the optimal $J \geqslant 0$. Taking $j_0 = 0$ — i.e. $|\bar{\Lambda}_{j_0}| \leqslant \lambda$ — and $J = \lceil \frac{1}{d} \log_2(T) \rceil$ entails the desired bound in the 3 cases $d < 2s$, $d = 2s$ and $d > 2s$.

**Remark.** In the preceding proof we showed that a single (linear) global resolution level $J = \lceil \frac{1}{d} \log_2 T \rceil$ suffices to attain the minimax regret for Hölder-smooth competitors, in contrast to general competitors in $B_{pq}^s(\mathcal{X})$, which require a *nonlinear* mechanism (see Appendix A). Nevertheless, even in the Hölder-smooth case one may take a larger level $J = \lceil \frac{S}{d\varepsilon} \log_2 T \rceil \geqslant \lceil \frac{1}{d} \log_2 T \rceil$ as in Theorem 1. In the analysis, set the oracle coefficients $\beta_{j,k} = 0$ for levels $j > \lceil \frac{1}{d} \log_2 T \rceil$; the estimation regret, combined with Assumption 1, reduces to a sum over the remaining nonzero coefficients—namely, those in the linear part—and leads to the same rates.

# C  Proof of Theorem 2

The proof uses a first key result that we state and prove right after.

**Theorem 3** (Local regret over Besov spaces). *Let $T \geqslant 1, 1 \leqslant p, q \leqslant \infty, s > \frac{d}{p}$ and $f \in B_{pq}^s$. Under the same assumptions of Theorem 1 and Assumption 2, Algorithm 2 with $\|f\|_\infty \leqslant B$ has regret*

$$R_T(f) \lesssim G \inf_{\mathcal{T}'} \left\{ \sum_{n \in \mathcal{L}(\mathcal{T}')} B\sqrt{|T_n|} + \|f\|_{s_n} \cdot 2^{-l(n)s_n} \cdot |T_n|^{r(s_n,p)} \right\},$$

*and if $(\ell_t)$ are exp-concave:*

$$R_T(f) \lesssim G \inf_{\mathcal{T}'} \left\{ B|\mathcal{L}(\mathcal{T}')| + \sum_{n \in \mathcal{L}(\mathcal{T}')} \|f\|_{s_n} \cdot 2^{-l(n)s_n} \cdot |T_n|^{r(s_n,p)} \right\},$$

*where $\lesssim$ hides logarithmic factors in $T$, and constants independent of $f$ or $T$, $\mathcal{L}(\mathcal{T}')$ denotes the set of leaves in a pruning $\mathcal{T}' \subset \mathcal{T}$, $\|f\|_{s_n}$ are local Besov norms, $l(n)$ is the level of node $n \in \mathcal{L}(\mathcal{T}')$, and the local rate exponent is given by*

$$r(s_n, p) = \begin{cases} \frac{1}{2} & \text{if } s_n \geqslant \frac{d}{2} \text{ or } p < 2, \\ 1 - \frac{s_n}{d} & \text{otherwise.} \end{cases}$$

**Remark.**  Theorem 3 holds for any pruning $\mathcal{T}'$ of $\mathcal{T}$. In particular, our procedure effectively competes against the best pruning with respect to the profile of the competitor $f$. Intuitively, Algorithm 2 achieves a spatial trade-off over the input space: it can refine locally by going deeper with high $l(n)$ at the cost of increasing the number of leaves $|\mathcal{L}(\mathcal{T}')|$, while remaining coarser and less accurate in other regions, with fewer leaves to compete against. In particular, when applying the result to a specific pruning, we show in Theorem 4 that Algorithm 2 achieves minimax-optimal (local) regret when facing exp-concave losses.

***Proof of Theorem 3.***  Let $1 \leqslant p, q \leqslant \infty, s > \frac{d}{p}, f \in B_{pq}^s$ such that $B := \|f\|_\infty < \infty$ - this is possible since $f$ is continuous over $\mathcal{X}$ with the condition $s > d/p$ and embedding of $B_{pq}^s$ in $L^\infty$.

**Grid for scaling coefficients at starting scale $j_0$.**  Observe that

$$|\alpha_{j_0,k}| = |\langle f, \phi_{j_0,k}\rangle| \leqslant \|f\|_\infty \cdot \|\phi_{j_0,k}\|_1 \leqslant 2^{-\frac{j_0 d}{2}} \|\phi\|_1 B.$$

Let $\varepsilon_{j_0} > 0$. We define the regular grid $\mathcal{A}_{j_0}$ of $\varepsilon_{j_0}$-precision, used to learn the scaling coefficients at level $j_0$, denoted $(\alpha_{j_0,k})$, with

$$|\mathcal{A}_{j_0}| = \left\lceil 2^{-j_0 \frac{d}{2}} 2B \|\phi\|_1 \varepsilon_{j_0}^{-1} \right\rceil$$

points, regularly spaced in the interval $[-B2^{-j_0 \frac{d}{2}} \|\phi\|_1, \ B2^{-j_0 \frac{d}{2}} \|\phi\|_1]$. In the following, we will use a local grid $\mathcal{A}_{l(n)}$ to learn scaling coefficient $\alpha_{n,k}$ at a scale $j_0 = l(n)$ locally over the space $\mathcal{X}$. In particular, we will carefully set the local precision $\varepsilon_{l(n)}$ to handle regret terms.

**Definition of the oracle associated to a pruning.**  Let $\mathcal{T}'$ be some pruning of $\mathcal{T}$ and $\mathcal{P}(\mathcal{T}') = (\mathcal{X}_n)_{n \in \mathcal{L}(\mathcal{T}')}$ be the associated partition of $\mathcal{X}$. Let $\mathcal{A}_{l(n)}$ denote the grid of precision $\varepsilon_{l(n)}$ as described above. We define the prediction function of pruning $\mathcal{T}'$, at any time $t \geqslant 1$

$$\hat{f}_{\mathcal{T}',t}(x) = \sum_{n \in \mathcal{L}(\mathcal{T}')} [\hat{f}_{n,\mathbf{a}_n,t}(x)]_B, \qquad x \in \mathcal{X},$$

where each $\hat{f}_{n,\mathbf{a}_n,t}$ is a sequential predictor of type (3), with starting scale $j_0 = l(n)$, restricted to $\mathcal{X}_n$, and initialized at the oracle scaling coefficients

$$\mathbf{a}_n = (a_{n,k})_{k \in \bar{\Lambda}_{j_0,n}} = \underset{\mathbf{a} \in \mathcal{A}_{l(n)}}{\arg\min} \|\mathbf{a} - \boldsymbol{\alpha}_{j_0,n}\|_\infty,$$

that is, the best approximating vector $\mathbf{a}$ in the grid $\mathcal{A}_{l(n)}$ for the subset of scaling coefficients $\boldsymbol{\alpha}_{j_0,n,k}, k \in \bar{\Lambda}_{j_0,n}$, whose basis functions $\phi_{j_0,k}$ are supported on $\mathcal{X}_n$. For simplicity, we slightly abuse notation by writing $\mathbf{a} \in \mathcal{A}_{l(n)}$, treating the grid as a tensor grid of the same dimension as $\mathbf{a}$. In particular, the number of coefficients in $\bar{\Lambda}_{j_0}$ whose supports intersect $\mathcal{X}_n$ satisfies

$$|\bar{\Lambda}_{j_0,n}| \leqslant |\bar{\Lambda}_{j_0}| 2^{-l(n)d} \leqslant \lambda,$$

since $l(n) = j_0$.

**Decomposition of regret.** We have a decomposition of regret as:

$$\text{Reg}_T(f) = \underbrace{\sum_{t=1}^{T} \ell_t(\hat{f}_t(x_t)) - \ell_t(\hat{f}_{\mathcal{T}',t}(x_t))}_{=:R_1} + \underbrace{\sum_{t=1}^{T} \ell_t(\hat{f}_{\mathcal{T}',t}(x_t)) - \ell_t(f(x_t))}_{=:R_2}, \qquad (36)$$

$R_1$ is the regret related to the estimation error of the expert-aggregation algorithm compared to some oracle partition $\mathcal{P}(\mathcal{T}')$ associated to $\mathcal{T}'$, i.e. the error the algorithm commits while aiming the oracle partition $\mathcal{P}(\mathcal{T}')$. On the other hand, $R_2$ is related to the error of the model predicting over subregions in $\mathcal{P}(\mathcal{T}')$, against some function $f \in B_{pq}^s$ and corresponds to the (localised) regret discussed in Theorem 1.

**Step 1: Upper-bounding $R_2$ as local regrets.** Recall that $\mathcal{P}(\mathcal{T}')$ form a partition of $\mathcal{X}$. Hence, for any $x_t \in \mathcal{X}$, the prediction at time $t$ is $\hat{f}_{\mathcal{T}',t}(x_t) = [\hat{f}_{j_0,n,\mathbf{a}_n,t}(x_t)]_B$ with $n \in \mathcal{N}(\mathcal{T}')$ the unique node such that $x_t \in \mathcal{X}_n$ at time $t$. Then, $R_2$ can be written as follows:

$$\begin{aligned}
R_2 &= \sum_{t=1}^{T} \sum_{n \in \mathcal{L}(\mathcal{T}')} (\ell_t(\hat{f}_{\mathcal{T}',t}(x_t)) - \ell_t(f(x_t))) \mathbb{1}_{x_t \in \mathcal{X}_n} \\
&= \sum_{n \in \mathcal{L}(\mathcal{T}')} \sum_{t \in T_n} \ell_t([\hat{f}_{n,\mathbf{a}_n,t}(x_t)]_B) - \ell_t(f(x_t)) \\
&\leqslant \sum_{n \in \mathcal{L}(\mathcal{T}')} \sum_{t \in T_n} \ell_t(\hat{f}_{n,\mathbf{a}_n,t}(x_t)) - \ell_t(f(x_t)), \qquad (37)
\end{aligned}$$

where we set $T_n = \{1 \leqslant t \leqslant T : x_t \in \mathcal{X}_n\}, \mathcal{X}_n \subset \mathcal{X}, n \in \mathcal{L}(\mathcal{T}')$ and (37) is because $[\hat{f}_{n,\mathbf{a}_n,t}]_B \leqslant \hat{f}_{n,\mathbf{a}_n,t}$ and $\ell_t$ is convex and has minimum in $[-B, B]$ with $B \geqslant \|f\|_\infty$.

The decomposition in (37) represents a sum of *local* error approximations of the function $f$ over the partition $\mathcal{P}(\mathcal{T}')$, using predictors $\hat{f}_{n,\mathbf{a}_n}, n \in \mathcal{L}(\mathcal{T}')$. Recall that for every $n \in \mathcal{L}(\mathcal{T}'), \hat{f}_{n,\mathbf{a}_n}$ is a prediction function associated to a wavelet decomposition (3), where the scaling coefficients start at $\mathbf{a}_n$ over $\mathcal{X}_n$ and with $j_0 = \mathrm{l}(n)$. In proof of Theorem 1 (Appendix A) we showed that any wavelet decomposition adapts to any regularity via $\|f\|_{B_{pq}^s}, s$ of $f$. Thus, the approximation error of $\hat{f}_{j_0,n,\mathbf{a}_n}$ with respect to $f$ remains similar to that in (27), but now with regard to a Besov function with local smoothness $s_n$ and norm $\|f\|_{s_n} := \|f\|_{B_{pq}^{s_n}(\mathcal{X}_n)}$ over $\mathcal{X}_n$ - see (10). Specifically, from (37), (23), (27), we get (without applying Hölder's inequality on the scaling coefficients):

$$\begin{aligned}
R_2 \leqslant \sum_{n \in \mathcal{L}(\mathcal{T}')} \Bigg[ & G\|\phi\|_\infty \sum_{k \in \bar{\Lambda}_{j_0,n}} |\alpha_{j_0,k} - a_{n,k}| \big(C_1 \sqrt{|T_n|} + C_2 2^{\frac{\mathrm{l}(n)d}{2}}\big) \\
& + \underbrace{\sum_{j=\mathrm{l}(n)}^{\mathrm{l}(n)+J_n^*} \lambda \|\boldsymbol{\beta}_j\|_p 2^{dj(\frac{1}{2}-\frac{1}{p})_+} \Big( C_1 \sqrt{\textstyle\sum_{k \in \Lambda_{j,n}} \sum_{t=1}^T |g_{j,k,t}|^2} + C_2 G \|\psi\|_\infty 2^{\frac{jd}{2}} 2^{dj(1-\frac{1}{p})} \Big)}_{\text{estimation error on wavelet coefficients as in (23)}} \\
& + \underbrace{2^{-(\mathrm{l}(n)+J_n^*)(s_n - \frac{d}{p})} |\Lambda_n^*|^{(\frac{1}{2}-\frac{1}{p})_+}}_{\text{estimation error on nonlinear wavelet coefficients}} \\
& + \underbrace{C_5 G \|f\|_{s_n} \big( 2^{-(\mathrm{l}(n)+J_n^*)(s_n - \frac{d}{p})} |\Lambda_n^*|^{-\frac{1}{p}} |T_n| + 2^{-(\mathrm{l}(n)+J_n)(s_n - \frac{d}{p})} |T_n| \big)}_{\text{approximation error (27) over } \mathcal{X}_n \text{ at scale } j_0 + J_n} \Bigg], \quad (38)
\end{aligned}$$

with $C_1, C_2$ as in Assumption 1 and $C_5$ a constant that can be deduced from (27) and $j_0 = \mathrm{l}(n)$ for each $n \in \mathcal{L}(\mathcal{T}')$. In particular, by definition of $\mathbf{a}_n = \arg\min_{\mathbf{a} \in \mathcal{A}_{\mathrm{l}(n)}} \|\boldsymbol{\alpha}_{j_0,n} - \mathbf{a}\|_\infty$, and given that $\mathcal{A}_{\mathrm{l}(n)}$ is a grid with precision $\varepsilon_{\mathrm{l}(n)} > 0$, one has

$$|\alpha_{j_0,k} - a_{n,k}| \leqslant \frac{\varepsilon_{\mathrm{l}(n)}}{2} \quad \text{for every } k \in \bar{\Lambda}_{j_0,n}.$$

From (38), one can bound the absolute values of the scaling terms by $\varepsilon_{\mathrm{l}(n)}/2$ and using $|\bar{\Lambda}_{j_0,n}| \leqslant \lambda$. For every $n \in \mathcal{L}(\mathcal{T}')$, let

$$\varepsilon_{\mathrm{l}(n)} = B \big( 2^{\frac{\mathrm{l}(n)d}{2}} \sqrt{T} \big)^{-1}$$

that gives for every $n \in \mathcal{L}(\mathcal{T}')$

$$\sum_{k \in \bar{\Lambda}_{j_0,n}} \frac{\varepsilon_{l(n)}}{2} \left( C_1 \sqrt{|T_n|} + C_2 2^{\frac{l(n)d}{2}} \right) \leqslant \frac{\lambda}{2} B \left( C_1 2^{-\frac{l(n)d}{2}} + C_2 T^{-\frac{1}{2}} \right),$$

where we used $\sqrt{|T_n|}/\sqrt{T} \leqslant 1$. Then, one can factorize the sum in $j$ and the approximation term by $2^{-l(n)s_n}$ over each $n \in \mathcal{L}(\mathcal{T}')$. Finally, applying Hölder's inequality over the sum in $j$ (see (21)) and following the same optimization steps in $J_n^*, J_n, |\Lambda_n^*|$ as in Proof of Theorem 1 we get, with $M$ defined as in (21):

$$R_2 \leqslant \lambda G M B |\mathcal{L}(\mathcal{T}')| \left( C_1 + C_2 T^{-\frac{1}{2}} \right)$$
$$+ \lambda G M \sum_{n \in \mathcal{L}(\mathcal{T}')} C_n \|f\|_{s_n} 2^{-l(n)s_n} \begin{cases} \sqrt{|T_n|} & \text{if } s_n \geqslant \frac{d}{2} \text{ or } p < 2 \\ |T_n|^{1-\frac{s_n}{d}} & \text{else,} \end{cases} \quad (39)$$

where $C_n = C_n(C_1, C_2, C_3, s_n, \psi, p)$ can be deduced from similar calculation as in (30), (31) and (32) and can include $\log T$ dependencies.

**Step 2: Upper-bounding the estimation error $R_1$.** $R_1$ is due to the error incurred by sequentially learning the prediction rule $\hat{f}_{\mathcal{T}'}$ associated with an oracle pruning $\mathcal{T}'$ of $\mathcal{T}$, along with the best scaling coefficients $(\mathbf{a}_n)_{n \in \mathcal{L}(\mathcal{T}')}$ selected from the grid $(\mathcal{A}_{l(n)})_{n \in \mathcal{L}(\mathcal{T}')}$.

Note that at each time $t$, only a subset of nodes in $\mathcal{T}$ are active and output predictions. Specifically, for any time $t \geqslant 1$, we define in Algorithm 2 the set of active experts at round $t$ as

$$\mathcal{E}_t = \{(n, \mathbf{a}_n) : x_t \in \mathcal{X}_n\}.$$

Moreover, we assume bounded gradients: for any time $t \geqslant 1$ and expert $e \in \mathcal{E}$,

$$|\nabla_{t,e}| = \left| \ell_t'(\hat{f}_t(x_t)) \cdot [\hat{f}_{e,t}(x_t)]_B \right| \leqslant GB,$$

which satisfies Assumption 2 with $\tilde{G} = BG$.

Using standard sleeping reduction, one can prove that, for any expert $(n, \mathbf{a}_n), n \in \mathcal{L}(\mathcal{T}'), t \geqslant 1$ - see Proof of Theorem 2 in [29] Eq. (31)-(35):

$$(\ell_t(\hat{f}_t(x_t)) - \ell_t(\hat{f}_{n,\mathbf{a}_n,t}(x_t))) \mathbb{1}_{x_t \in \mathcal{X}_n} \leqslant \ell_t'(\hat{f}_t(x_t))(\hat{f}_t(x_t) - \hat{f}_{n,\mathbf{a}_n,t}(x_t)) \mathbb{1}_{x_t \in \mathcal{X}_n} \quad \leftarrow \text{by convexity of } \ell_t$$
$$= (\nabla_t^\top \tilde{\mathbf{w}}_t - \nabla_{(n,\mathbf{a}_n),t}) \mathbb{1}_{x_t \in \mathcal{X}_n} \quad (40)$$
$$= \nabla_t^\top \mathbf{w}_t - \nabla_{(n,\mathbf{a}_n),t}. \quad (41)$$

Then, with $T_n = \{1 \leqslant t \leqslant T : x_t \in \mathcal{X}_n\}, n \in \mathcal{L}(\mathcal{T}')$:

$$R_1 = \sum_{t=1}^{T} \sum_{n \in \mathcal{L}(\mathcal{T}')} (\ell_t(\hat{f}_t(x_t)) - \ell_t(\hat{f}_{n,\mathbf{a}_n,t}(x_t))) \mathbb{1}_{x_t \in \mathcal{X}_n} \quad \leftarrow \{\mathcal{X}_n, n \in \mathcal{L}(\mathcal{T}')\} \text{ partition of } \mathcal{X}$$

$$\leqslant \sum_{n \in \mathcal{L}(\mathcal{T}')} \sum_{t=1}^{T} (\nabla_t^\top \mathbf{w}_t - \nabla_{(n,\mathbf{a}_n),t}) \quad \leftarrow \text{by (41)}$$

$$\leqslant \sum_{n \in \mathcal{L}(\mathcal{T}')} \left( C_3 \sqrt{\log(|\mathcal{E}|)} \sqrt{\sum_{t=1}^{T} (\nabla_t^\top \mathbf{w}_t - \nabla_{(n,\mathbf{a}_n),t})^2} + C_4 \tilde{G} \right) \quad \leftarrow \text{by Assumption 2}$$

$$= C_4 B G |\mathcal{L}(\mathcal{T}')| + C_3 \sqrt{\log(|\mathcal{E}|)} \sum_{n \in \mathcal{L}(\mathcal{T}')} \sqrt{\sum_{t \in T_n} (\nabla_t^\top \mathbf{w}_t - \nabla_{(n,\mathbf{a}_n),t})^2}, \quad (42)$$

where the last equality holds because for any $n \in \mathcal{L}(\mathcal{T}'), \nabla_t^\top \mathbf{w}_t - \nabla_{(n,\mathbf{a}_n),t} = 0$ if $x_t \notin \mathcal{X}_n$ and $\tilde{G} = BG$.

The proof goes on with two different cases depending on the losses' convex properties:

- *Case 1: $(\ell_t)_{1 \leqslant t \leqslant T}$ convex.*
  Observe that at any time $t \in [T], \|\nabla_t\|_\infty \leqslant \tilde{G}$ and $\|\mathbf{w}_t\|_1 = 1$, which gives $|\nabla_t^\top \mathbf{w}_t| \leqslant \tilde{G} = BG$. Then, from (42)

$$R_1 \leqslant C_4 B G |\mathcal{L}(\mathcal{T}')| + 2 C_3 \sqrt{\log(|\mathcal{E}|)} B G \sum_{n \in \mathcal{L}(\mathcal{T}')} \sqrt{|T_n|} \quad (43)$$

In case of convex losses, we finally have by (36), (39) and (43):

$$\text{Reg}_T(f) \leqslant 2C_3 BG\sqrt{\log\left(|\mathcal{E}|\right)} \sum_{n\in\mathcal{L}(\mathcal{T}')} \sqrt{|T_n|} + \lambda GBM|\mathcal{L}(\mathcal{T}')|(C_1 + C_2 T^{-\frac{1}{2}})$$

$$+ \lambda GM \sum_{n\in\mathcal{L}(\mathcal{T}')} C_n \|f\|_{s_n} 2^{-\mathbf{l}(n)s_n} \begin{cases} \sqrt{|T_n|} & \text{if } s_n \geqslant \frac{d}{2} \text{ or } p < 2, \\ |T_n|^{1-\frac{s_n}{d}} & \text{else,} \end{cases} \quad (44)$$

where $|\mathcal{E}| \leqslant |\mathcal{N}(\mathcal{T})|\big(2\|\phi\|_1 T^{\frac{1}{2}}\big)^\lambda$ since for every $n \in \mathcal{T}$ one has $|\bar{\Lambda}_{j_0,n}| \leqslant \lambda$ and

$$|\mathcal{A}_{\mathbf{l}(n)}| = \lceil 2B\|\phi\|_1/\varepsilon_{\mathbf{l}(n)} \rceil = \lceil 2\|\phi\|_1 T^{\frac{1}{2}} \rceil$$

by the choice of the precision $\varepsilon_{\mathbf{l}(n)} = B2^{-\frac{\mathbf{l}(n)d}{2}}T^{-\frac{1}{2}}$. In particular, the grids have a number of points that does not grow exponentially with $T$, making the construction computationally feasible. Finally, since (44) holds for all pruning $\mathcal{T}'$ of our main tree $\mathcal{T}$, one can take the infimum over all pruning to get the desired upper-bound.

- *Case 2:* $(\ell_t)_{1\leqslant t\leqslant T}$ $\eta$-*exp-concave.*
  If the sequence of loss functions $(\ell_t)$ is $\eta$-exp-concave for some $\eta > 0$, then thanks to [24, Lemma 4.3] we have for any $0 < \mu \leqslant \frac{1}{2}\min\{\frac{1}{G},\eta\}$ and all $t \geqslant 1, n \in \mathcal{L}(\mathcal{T}')$, using (41):

$$(\ell_t(\hat{f}_t(x_t)) - \ell_t(\hat{f}_{n,\mathbf{a}_n,t}(x_t)))\mathbb{1}_{x_t\in\mathcal{X}_n} \leqslant \nabla_t^\top \mathbf{w}_t - \nabla_{(n,\mathbf{a}_n),t} - \frac{\mu}{2}\big(\nabla_t^\top \mathbf{w}_t - \nabla_{(n,\mathbf{a}_n),t}\big)^2 \quad (45)$$

Summing (45) over $t \in [T]$ and $n \in \mathcal{L}(\mathcal{T}')$, we get:

$$R_1 \leqslant \sum_{n\in\mathcal{L}(\mathcal{T}')}\sum_{t\in T_n} \nabla_t^\top \mathbf{w}_t - \nabla_{(n,\mathbf{a}_n),t} - \frac{\mu}{2}\sum_{n\in\mathcal{L}(\mathcal{T}')}\sum_{t\in T_n}\big(\nabla_t^\top \mathbf{w}_t - \nabla_{(n,\mathbf{a}_n),t}\big)^2$$

$$\leqslant C_4\tilde{G}|\mathcal{L}(\mathcal{T}')| + \tilde{C}_3 \sum_{n\in\mathcal{L}(\mathcal{T}')}\sqrt{\sum_{t\in T_n}\big(\nabla_t^\top \mathbf{w}_t - \nabla_{(n,\mathbf{a}_n),t}\big)^2} - \frac{\mu}{2}\sum_{n\in\mathcal{L}(\mathcal{T}')}\sum_{t\in T_n}\big(\nabla_t^\top \mathbf{w}_t - \nabla_{(n,\mathbf{a}_n),t}\big)^2,$$

$$(46)$$

where last inequality is by (42) and we set $\tilde{C}_3 = C_3\sqrt{\log\left(|\mathcal{E}|\right)}$ and $\tilde{G} = BG$. Young's inequality gives, for any $\nu > 0$, the following upper-bound:

$$\sqrt{\sum_{t\in T_n}\big(\nabla_t^\top \mathbf{w}_t - \nabla_{(n,\mathbf{a}_n),t}\big)^2} \leqslant \frac{1}{2\nu} + \frac{\nu}{2}\sum_{t\in T_n}\big(\nabla_t^\top \mathbf{w}_t - \nabla_{(n,\mathbf{a}_n),t}\big)^2. \quad (47)$$

Finally, plugging (47) with $\nu = \mu/\tilde{C}_3 > 0$ in (46), we get

$$R_1 \leqslant C_4 BG|\mathcal{L}(\mathcal{T}')| + \tilde{C}_3 \sum_{n\in\mathcal{L}(\mathcal{T}')}\left(\frac{\tilde{C}_3}{2\mu} + \frac{\mu}{2\tilde{C}_3}\sum_{t\in T_n}\big(\nabla_t^\top \mathbf{w}_t - \nabla_{(n,\mathbf{a}_n),t}\big)^2\right)$$

$$- \frac{\mu}{2}\sum_{n\in\mathcal{L}(\mathcal{T}')}\sum_{t\in T_n}\big(\nabla_t^\top \mathbf{w}_t - \nabla_{(n,\mathbf{a}_n),t}\big)^2$$

$$= \left(\frac{C_3^2\log\left(|\mathcal{E}|\right)}{2\mu} + C_4 BG\right)|\mathcal{L}(\mathcal{T}')|, \quad (48)$$

again with $|\mathcal{E}| \leqslant |\mathcal{N}(\mathcal{T})|\big(2\|\phi\|_1 T^{\frac{1}{2}}\big)^\lambda$. Then, one can deduce the final bound from equations (36), (39) and (48) and taking the infimum over the prunings $\mathcal{T}'$.

**Worst case regret bound.** Note that since we assume that $\|f\|_\infty \leqslant B$, and that all local predictors $\hat{f}_e, e \in \mathcal{E}$ in Algorithm 2 are clipped in $[-B, B]$, we first have for any $x \in \mathcal{X}$,

$$|\hat{f}_t(x)| = \sum_{e\in\mathcal{E}_t} w_{e,t}|[\hat{f}_{e,t}(x)]_B| \leqslant B\sum_{e\in\mathcal{E}_t} w_{e,t} = B.$$

Thus,

$$
\begin{aligned}
\mathrm{Reg}_T(f) &= \sum_{t=1}^{T} \ell_t(\hat{f}_t(x_t)) - \ell_t(f(x_t)) \\
&\leqslant \sum_{t=1}^{T} G|\hat{f}_t(x_t) - f(x_t)| && \leftarrow \ell_t \text{ is } G\text{-Lipschitz} \\
&\leqslant G \sum_{t=1}^{T} |\hat{f}_t(x_t)| + |f(x_t)| \\
&= 2BGT && (49)
\end{aligned}
$$

$\square$

We now restate a complete version of Theorem 2 from the main text and provide its proof below.

**Theorem 4.** *Let $T \geqslant 1, 1 \leqslant p, q \leqslant \infty, s > \frac{d}{p}$. Let $f \in B_{pq}^s$ and $B \geqslant \|f\|_\infty$. Let $\mathcal{T}'$ be any pruning of $\mathcal{T}$, together with a collection of local smoothness indices $(s_n)_{n \in \mathcal{L}(\mathcal{T}')}$ defined as in (10) and local norms $\|f\|_{s_n}$. Then, under the same assumptions of Theorem 1 and Assumption 2, Algorithm 2 satisfies*

$$
R_T(f) \lesssim G \sum_{n \in \mathcal{L}(\mathcal{T}')} \left( B^{1 - \frac{d}{2s_n}} \left(2^{-\mathrm{l}(n)s_n} \|f\|_{s_n}\right)^{\frac{d}{2s_n}} \sqrt{|T_n|} \mathbb{1}_{s_n \geqslant \frac{d}{2}} \right.
$$

$$
\left. + \left(2^{-\mathrm{l}(n)s_n} \|f\|_{s_n} |T_n|^{1 - \frac{s_n}{d}}\right) \mathbb{1}_{s_n < \frac{d}{2}} + B\sqrt{|T_n|} \right)
$$

*and moreover we also have, if $(\ell_t)$ are exp-concave:*

$$
R_T(f) \lesssim G \sum_{n \in \mathcal{L}(\mathcal{T}')} \left( B^{1 - \frac{2d}{2s_n + d}} \left(2^{-\mathrm{l}(n)s_n} \|f\|_{s_n}\right)^{\frac{2d}{2s_n + d}} |T_n|^{\frac{d}{2s_n + d}} \mathbb{1}_{s_n \geqslant \frac{d}{2}} \right.
$$

$$
\left. + 2^{-\mathrm{l}(n)s_n} \|f\|_{s_n} |T_n|^{1 - \frac{s_n}{d}} \mathbb{1}_{s_n < \frac{d}{2}} + B \right),
$$

*where $\lesssim$ hides logarithmic factors in $T$, and constants independent of $f$ or $T$.*

***Proof of Theorem 4.***
Let $\mathcal{T}' \subset \mathcal{T}$ be some pruning of $\mathcal{T}$. We define $\mathcal{T}'_{\mathrm{ext}}$ the extension of $\mathcal{T}'$ such that all terminal nodes $n \in \mathcal{L}(\mathcal{T}')$ is extended with a tree $\mathcal{T}'_n$ of depth $h_n \in \mathbb{N}$. In particular, for any $n' \in \mathcal{L}(\mathcal{T}'_{\mathrm{ext}}), \mathrm{l}(n') = \mathrm{l}(n) + h_n$ with $n \in \mathcal{L}(\mathcal{T}')$. See Figure 4 for an illustrative example.

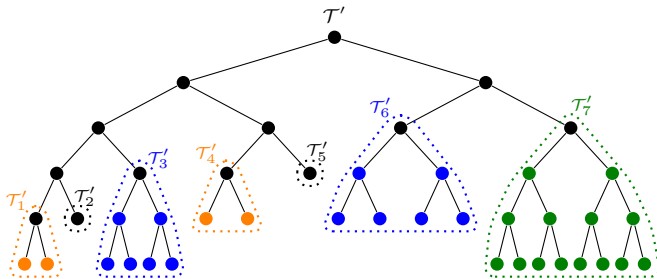

Figure 4: Example of an extended tree $\mathcal{T}'_{\mathrm{ext}} = \mathcal{T}' \cup \mathcal{T}'_1 \cup \cdots \cup \mathcal{T}'_7$, formed by a subtree $\mathcal{T}'$ (black nodes) and its extensions (colored nodes). Each dotted set corresponds to a subtree $\mathcal{T}'_n$, rooted at a leaf $n \in \mathcal{L}(\mathcal{T}')$ and extended to depth $h_n$. The depths vary: $h_2 = h_5 = 0$ (black boxes), $h_1 = h_4 = 1$ (orange), $h_3 = h_6 = 2$ (blue), and $h_7 = 3$ (green). The leaves of $\mathcal{T}'_{\mathrm{ext}}$ appear at different levels depending on the values of $(h_n)$ and the level $\mathrm{l}(n)$ of the leaves $n \in \mathcal{L}(\mathcal{T}') = \{1, 2, 3, 4, 5, 6, 7\}$.

Observe that the total number of leaves in the extended pruning $\mathcal{T}'_{\mathrm{ext}}$ is

$$
|\mathcal{L}(\mathcal{T}'_{\mathrm{ext}})| = \sum_{n \in \mathcal{L}(\mathcal{T}')} |\mathcal{L}(\mathcal{T}'_n)|. \tag{50}
$$

Define
$$s_n := \min_{n' \in \mathcal{L}(\mathcal{T}'_{\text{ext}})} s_{n'}, \quad n \in \mathcal{N}(\mathcal{T}'). \tag{51}$$

Remark also that by definition of the Besov norm in (6) (and via the usual embedding $(p, q)$ fixed - see, e.g., [20]), one has for any $n \in \mathcal{L}(\mathcal{T}'), \|f\|_{s_n} \geqslant \|f\|_{s_{n'}}, n' \in \mathcal{L}(\mathcal{T}'_{\text{ext}})$. In particular, every tree extension $\mathcal{T}'_n$ at node $n \in \mathcal{L}(\mathcal{T}')$ has $|\mathcal{L}(\mathcal{T}'_n)| = 2^{h_n d}$ leaves.

**Case $(\ell_t)$ convex.** Applying Theorem 3 in the convex case on the extended pruning $\mathcal{T}'_{\text{ext}}$, gives

$$R_T(f) \leqslant CG \sum_{n' \in \mathcal{L}(\mathcal{T}'_{\text{ext}})} \left( B\sqrt{|T_{n'}|} + \|f\|_{s_{n'}} \cdot 2^{-1(n')s_{n'}} \cdot |T_{n'}|^{r_{n'}} \right) \tag{52}$$

with $C$ some constant that hides $\log T$ terms that can change from an inequality to another and $r_{n'} \in \{\frac{1}{2}, 1 - \frac{s_{n'}}{d}\}$ is the local rate described in Theorem 3. Note that by (51) one has $r'_n \leqslant r_n, n' \in \mathcal{L}(\mathcal{T}'_n)$. Recall that for every $n' \in \mathcal{L}(\mathcal{T}'_{\text{ext}}), l(n') = l(n) + h_n$ for $n \in \mathcal{L}(\mathcal{T}')$ and since every leaves in $\mathcal{L}(\mathcal{T}'_n)$ is partitioning each terminal node $n \in \mathcal{T}'$, one has by Jensen's inequality:

$$\sum_{n' \in \mathcal{L}(\mathcal{T}'_{\text{ext}})} \sqrt{|T_{n'}|} = \sum_{n \in \mathcal{L}(\mathcal{T}')} \sum_{n' \in \mathcal{L}(\mathcal{T}'_n)} \sqrt{|T_{n'}|} \leqslant \sum_{n \in \mathcal{L}(\mathcal{T}')} \sqrt{|\mathcal{L}(\mathcal{T}'_n)||T_n|}. \tag{53}$$

Then, by (51),(52) and (53) one gets (with $r_{n'} \leqslant r_n, n' \in \mathcal{L}(\mathcal{T}'_n)$)

$$R_T(f) \leqslant CG \sum_{n \in \mathcal{L}(\mathcal{T}')} \sum_{n' \in \mathcal{L}(\mathcal{T}'_n)} \left( B\sqrt{|T_{n'}|} + \|f\|_{s_n} 2^{-(l(n)+h_n)s_n} |T_{n'}|^{r_n} \right)$$

$$\leqslant CG \sum_{n \in \mathcal{L}(\mathcal{T}')} \left( B\sqrt{|\mathcal{L}(\mathcal{T}'_n)||T_n|} + \|f\|_{s_n} 2^{-(l(n)+h_n)s_n} \sum_{n' \in \mathcal{L}(\mathcal{T}'_n)} |T_{n'}|^{r_n} \right). \tag{54}$$

Further, applying Hölder's inequality over the sum over $n' \in \mathcal{L}(\mathcal{T}'_n)$ in (54) with $(1 - r_n) + r_n = 1$ ($r_n$ is constant over the sum in $n'$):

$$R_T(f) \leqslant CG \sum_{n \in \mathcal{L}(\mathcal{T}')} \left( B\sqrt{|\mathcal{L}(\mathcal{T}'_n)||T_n|} + \|f\|_{s_n} 2^{-1(n)s_n} 2^{-h_n s_n} |\mathcal{L}(\mathcal{T}'_n)|^{1-r_n} \left( \sum_{n' \in \mathcal{L}(\mathcal{T}'_n)} |T_{n'}| \right)^{r_n} \right) \tag{55}$$

$$= CG \sum_{n \in \mathcal{L}(\mathcal{T}')} \left( B\sqrt{|T_n| 2^{h_n d}} + \|f\|_{s_n} 2^{-1(n)s_n} 2^{dh_n(1 - \frac{s_n}{d} - r_n)} |T_n|^{r_n} \right)$$

where we used

$$\sum_{n' \in \mathcal{L}(\mathcal{T}'_n)} |T_{n'}| = |T_n| \quad \text{and} \quad 2^{-h_n s_n} |\mathcal{L}(\mathcal{T}'_n)|^{1-r_n} = 2^{-h_n s_n} (2^{dh_n})^{1-r_n} = 2^{dh_n(1 - \frac{s_n}{d} - r_n)}.$$

Define the local regrets under the sum over $n \in \mathcal{N}(\mathcal{T}')$ by

$$R_n(f) := B\sqrt{|T_n| 2^{h_n d}} + \|f\|_{s_n} 2^{-1(n)s_n} 2^{dh_n(1 - \frac{s_n}{d} - r_n)} |T_n|^{r_n},$$

that we now want to optimize in $h_n \in \mathbb{N}$. This leads to two different cases depending on the values of the local exponent $r_n$, defined in Theorem 3.

- *Case $s_n \geqslant \frac{d}{2}$ or $p < 2$: $r_n = \frac{1}{2}$*
  The local regret grows as:
  $$B\sqrt{|T_n| 2^{h_n d}} + \|f\|_{s_n} 2^{-1(n)s_n} 2^{-h_n(s_n - \frac{d}{2})} \sqrt{|T_n|}.$$
  Therefore, setting $h_n = \max\left\{0, \left\lceil \frac{1}{s_n} \log_2 \left(2^{-1(n)s_n} \|f\|_{s_n} B^{-1}\right) \right\rceil\right\}$ this entails
  $$R_n(f) \leqslant C \max\{B, B^{1 - \frac{d}{2s_n}} (2^{-1(n)s_n} \|f\|_{s_n})^{\frac{d}{2s_n}}\} \sqrt{|T_n|}$$

- *Case $s_n < \frac{d}{2}$: $r_n = 1 - \frac{s_n}{d}$*
  The local regret grows as:
  $$R_n(f) = B\sqrt{|T_n| 2^{h_n d}} + \|f\|_{s_n} 2^{-1(n)s_n} |T_n|^{1 - \frac{s_n}{d}},$$
  and the best choice is $h_n = 0$ in this case that entails
  $$R_n(f) = B\sqrt{|T_n|} + \|f\|_{s_n} 2^{-1(n)s_n} |T_n|^{1 - \frac{s_n}{d}}$$

Finally, in the case $(\ell_t)$ are convex losses, we deduce that the regret is upper bounded as

$$R_T(f) \leqslant CG \sum_{n \in \mathcal{L}(\mathcal{T}')} R_n(f)$$

$$\leqslant CG \sum_{n \in \mathcal{L}(\mathcal{T}')} \left( \left( \max\{B, B^{1-\frac{d}{2s_n}} (2^{-1(n)s_n} \|f\|_{s_n})^{\frac{d}{2s_n}}\} \sqrt{|T_n|} \right) \mathbb{1}_{s_n \geqslant \frac{d}{2}} \right.$$

$$\left. + \left( B\sqrt{T_n} + 2^{-1(n)s_n} \|f\|_{s_n} |T_n|^{1-\frac{s_n}{d}} \right) \mathbb{1}_{s_n < \frac{d}{2}} \right) \tag{56}$$

**Case $(\ell_t)$ exp-concave.** Applying Theorem 3 in the exp-concave case on the extended pruning $\mathcal{T}'_{\text{ext}}$, gives

$$R_T(f) \leqslant CG \left( B|\mathcal{L}(\mathcal{T}'_{\text{ext}})| + \sum_{n' \in \mathcal{L}(\mathcal{T}'_{\text{ext}})} \|f\|_{s_{n'}} \cdot 2^{-1(n')s_{n'}} \cdot |T_{n'}|^{r_{n'}} \right) \tag{57}$$

with $C$ some constant that hides $\log T$ terms and that can change from an inequality to another and $r_{n'} \in \{\frac{1}{2}, 1 - \frac{s_{n'}}{d}\}$ is the local rate described in Theorem 3. Note that $|\mathcal{L}(\mathcal{T}'_{\text{ext}})| = \sum_{n \in \mathcal{L}(\mathcal{T}')} |\mathcal{L}(\mathcal{T}'_n)|$ and again for every $n' \in \mathcal{L}(\mathcal{T}'_{\text{ext}}), 1(n') = 1(n) + h_n$ for $n \in \mathcal{L}(\mathcal{T}')$ and $r_{n'} \leqslant r_n, n' \in \mathcal{L}(\mathcal{T}'_n)$. We get

$$R_T(f) \leqslant CG \sum_{n \in \mathcal{L}(\mathcal{T}')} \left( B|\mathcal{L}(\mathcal{T}'_n)| + \|f\|_{s_n} 2^{-(1(n)+h_n)s_n} \sum_{n' \in \mathcal{L}(\mathcal{T}'_n)} |T_{n'}|^{r_n} \right). \tag{58}$$

Using $|\mathcal{L}(\mathcal{T}'_n)| = 2^{h_n d}$ and applying Hölder's inequality over the sum over $n' \in \mathcal{L}(\mathcal{T}'_n)$ in (58) with $(1 - r_n) + r_n = 1$ as in (55) entails

$$R_T(f) \leqslant CG \sum_{n \in \mathcal{L}(\mathcal{T}')} \left( B2^{h_n d} + \|f\|_{s_n} 2^{-1(n)s_n} 2^{dh_n(1-\frac{s_n}{d}-r_n)} |T_n|^{r_n} \right).$$

Again, we define the local regrets under the sum over $n \in \mathcal{N}(\mathcal{T}')$ as

$$R_n(f) := B2^{h_n d} + \|f\|_{s_n} 2^{-1(n)s_n} 2^{dh_n(1-\frac{s_n}{d}-r_n)} |T_n|^{r_n},$$

that we optimize in $h_n \in \mathbb{N}$. The cases are the same as for the convex case, according to the values of the local exponent $r_n$, defined in Theorem 3.

- *Case $s_n \geqslant \frac{d}{2}$ or $p < 2$: $r_n = \frac{1}{2}$*
  The local regret grows as

  $$R_n(f) = B2^{h_n d} + \|f\|_{s_n} 2^{-1(n)s_n} 2^{-h_n(s_n - \frac{d}{2})} \sqrt{|T_n|}.$$

  Afterwards, optimizing in $h_n$ such that

  $$B2^{h_n d} = 2^{-1(n)s_n} \|f\|_{s_n} 2^{-h_n(s_n - \frac{d}{2})} \sqrt{|T_n|}$$

  leads to $h_n = \max\left\{0, \lceil \frac{1}{2s_n+d} \log_2 \left( (B^{-1} 2^{-1(n)s_n} \|f\|_{s_n})^2 |T_n| \right) \rceil \right\}$, that entails

  $$R_n(f) \leqslant C \max\left\{ B, B^{1-\frac{2d}{2s_n+d}} \left( 2^{-1(n)s_n} \|f\|_{s_n} \right)^{\frac{2d}{2s_n+d}} |T_n|^{\frac{d}{2s_n+d}} \right\}$$

- *Case $s_n < \frac{d}{2}$: $r_n = 1 - \frac{s_n}{d}$*
  The local regret grows as

  $$R_n(f) = B2^{h_n d} + 2^{-1(n)s_n} \|f\|_{s_n} |T_n|^{1-\frac{s_n}{d}},$$

  and the best choice is $h_n = 0$ which entails

  $$R_n(f) = B + 2^{-1(n)s_n} \|f\|_{s_n} |T_n|^{1-\frac{s_n}{d}},$$

Finally, with $(\ell_t)$ exp-concave losses, the regret is bounded as

$$R_T(f) \leqslant CG \sum_{n \in \mathcal{L}(\mathcal{T}')} R_n(f)$$

$$\leqslant CG \sum_{n \in \mathcal{L}(\mathcal{T}')} \left( \max\left\{ B, B^{1-\frac{2d}{2s_n+d}} \left( 2^{-1(n)s_n} \|f\|_{s_n} \right)^{\frac{2d}{2s_n+d}} |T_n|^{\frac{d}{2s_n+d}} \right\} \mathbb{1}_{s_n \geqslant \frac{d}{2}} \right. \tag{59}$$

$$\left. + \left( B + 2^{-1(n)s_n} \|f\|_{s_n} |T_n|^{1-\frac{s_n}{d}} \right) \mathbb{1}_{s_n < \frac{d}{2}} \right). \tag{60}$$

*Remark.* Taking $\mathcal{T}'$ as the pruning associated to the root, this entails $O(T^{\frac{d}{2s+d}}) = O(T^{1-\frac{2s}{2s+d}})$ which is minimax-optimal for this case - see [35].

$\square$

# D   Discussion on the unbounded case: $s < \frac{d}{p}$

As in most previous works in statistical learning, this paper primarily considers competitive functions $f \in B_{pq}^s(\mathcal{X})$ with $s > \frac{d}{p}$, which ensures that $f \in L^\infty(\mathcal{X})$ with $\|f\|_\infty < \infty$.

A natural question is whether our Algorithm 1 remains competitive - that is, achieves sublinear regret - in the more challenging regime where $s < \frac{d}{p}$. Indeed, in the case $s < \frac{d}{p}$, prediction rules may no longer be bounded in sup-norm. For example, the function $f(x) = x^{-1/2}\mathbf{1}_{x \in (0,1]}$ belongs to $L^p([0,1])$ for $p < 2$ but not to $L^\infty([0,1])$, illustrating the type of singularity permitted when $s < \frac{d}{p}$. In such cases, the boundedness condition $\|f_J - f\|_\infty < \infty$ required in (24) may fail, where $f_J$ denotes the truncated $J$-level wavelet expansion defined in (2). Nevertheless, we discuss how Algorithm 1 can still offer performance guarantees in certain settings, particularly when the input data $\{x_t\}$ are well distributed over $\mathcal{X}$.

Indeed, by Hölder's inequality, (24) can be upper bounded as:

$$\sum_{t=1}^{T} |f_J(x_t) - f(x_t)| \leqslant T\left(\frac{1}{T}\sum_{t=1}^{T} |f_J(x_t) - f(x_t)|^p\right)^{\frac{1}{p}}, \tag{61}$$

where the sum on the right-hand side defines an empirical $\ell^p$ semi-metric over the input data $\{x_t\}_{t=1}^T$, denoted:

$$d_T^p(f_J, f) := \left(\frac{1}{T}\sum_{t=1}^{T} |f_J(x_t) - f(x_t)|^p\right)^{\frac{1}{p}}.$$

The upper bound (61) suggests that tighter control may be obtained by focusing on the empirical norm $d_T^p(f_J, f)$ rather than on the sup-norm, which may not be finite.

**First case: the empirical semi-norm $d_T^p$ approximates the $L^p$ norm.**   Assume that the semi-norm $d_T^p(f_J, f)$ is close to the true $L^p$ norm $\|f_J - f\|_{L^p}$. Such an equivalence is expected when the data $\{x_t\}$ are well distributed over $\mathcal{X}$, for example when $x_t \sim \mathcal{U}(\mathcal{X})$ i.i.d., or when $x_t$ are equally spaced, such as $x_t = \frac{t}{T}$ for $t = 1, \ldots, T$. By the law of large numbers or standard concentration arguments, one typically has $d_T^p(f_J, f) \approx \|f_J - f\|_{L^p}$ in expectation or with high probability.

Classical approximation results (e.g., [20, Prop. 4.3.8]) then yield $\|f_J - f\|_{L^p} \lesssim 2^{-Js}$ for $f \in B_{pq}^s(\mathcal{X}) \subset L^p(\mathcal{X})$. Optimizing over $J$ to balance estimation and approximation regrets leads to a regret bound of $O(T^{1-\frac{s}{d}})$ - see the proof of Theorem A, last case $\beta < 0$. This regret is sublinear as soon as $s > 0$ and becomes linear when $s = 0$, as is typical for $f \in L^p$.

Nevertheless, minimax analysis from [34, 35] shows that a regret of $O(T^{1-1/p})$ is possible, which improves upon our bound whenever $s < \frac{d}{p}$. Whether a constructive algorithm achieving this minimax regret exists in the regime $s < \frac{d}{p}$ remains, to the best of our knowledge, an open and interesting question.

**Second case: the semi-norm $d_T^p$ fails to approximate the $L^p$ norm.**   If the data points $\{x_t\}$ are concentrated near singularities (e.g., near 0 in the example above), the empirical norm $d_T^p(f_J, f)$ can differ significantly from the true norm $\|f_J - f\|_{L^p}$, making the latter less informative in practice.

In such adversarial or non-uniform settings, it seems preferable to control the empirical norm $d_T^p(f_J, f)$ directly, as it more accurately reflects the distribution of the observed data. Addressing this challenge may require adaptive sampling strategies, localization techniques, or alternative norms that account for the geometry or density of the input distribution.

# E   Review of multi-resolution analysis

In this section we present some of the basic ingredients of wavelet theory. Let's assume we have a multivariate function $f : \mathbb{R}^d \to \mathbb{R}$.

**Definition 1** (Scaling function). *We say that a function $\phi \in L^2(\mathbb{R}^d)$ is the scaling function of a multiresolution analysis (MRA) if it satisfies the following conditions:*

1. *the family*
$$\{x \mapsto \phi(x - k) = \textstyle\prod_{i=1}^d \phi(x_i - n_i) : k \in \mathbb{Z}^d\}$$
   *is an ortho-normal basis, that is $\langle \phi(\cdot - k), \phi(\cdot - n) \rangle = \delta_{k,n}$;*

2. *the linear spaces*
$$V_0 = \left\{ f = \textstyle\sum_{k \in \mathbb{Z}^d} c_k \phi(\cdot - k), (c_k) : \textstyle\sum_{k \in \mathbb{Z}^d} c_k^2 < \infty \right\}, \ldots, V_j = \{h = f(2^j \cdot) : f \in V_0\}, \ldots,$$
   *are nested, i.e. $V_{j-1} \subset V_j$ for all $j \geqslant 0$.*

We note that under these two conditions, it is immediate that the functions
$$\{\phi_{j,k} = 2^{dj/2} \phi(2^j \cdot -k), k \in \mathbb{Z}^d\}$$

form an ortho-normal basis of the space $V_j, j \in \mathbb{N}$. One can define the projection kernel of $f$ over $V_j$ (from here we also say kernel projection at scale or level $j$) as

$$K_j f(x) := \sum_{k \in \mathbb{Z}^d} \langle f, \phi_{j,k} \rangle \phi_{j,k}(x) = \int_{\mathbb{R}^d} K_j(x,y) f(y)\, dy, \tag{62}$$

with $K_j(x,y) = \sum_{n \in \mathbb{Z}^d} \phi_{j,k}(x) \phi_{j,k}(y) = \sum_{k \in \mathbb{Z}^d} 2^{dj} \phi(2^j x - n) \phi(2^j y - n)$ (which is not of convolution type) but has comparable approximation properties that we detail after.

**Incremental construction via wavelets.**   Since the spaces $(V_j)$ are nested, one can define nontrivial subspaces as the orthogonal complements $W_j := V_{j+1} \ominus V_j$. We can then telescope these orthogonal complements to see that each space $V_j, j \geqslant j_0$ can be written as

$$V_j = V_{j_0} \oplus \left( \bigoplus_{l=j_0}^j W_l \right) \quad \text{for any } j_0 \in \mathbb{N}.$$

Let $\psi$ be a mother wavelet corresponding to the scaling function $\phi$. The associated wavelets are defined as follows: for $E = \{0,1\}^d \setminus \{0\}$, we set

$$\psi^\varepsilon(x) = \psi^{\varepsilon_1}(x_1) \cdots \psi^{\varepsilon_d}(x_d), \quad \psi_{j,n}^\varepsilon = 2^{jd/2} \psi^\varepsilon(2^j x - n), \quad j \geqslant 0, \quad n \in \mathbb{Z}^d,$$

where $\psi^0 = \phi, \psi^1 = \psi$. For each $j$, these functions form an orthonormal basis of $W_j$.
Analogously, one can now observe that for every $j \geqslant j_0$,

$$K_j f = K_{j_0} f + \sum_{l=j_0}^{j-1} (K_{l+1} f - K_l f), \tag{63}$$

where each increment in the sum can be written as

$$K_{j+1} f - K_j f = \sum_k \sum_\varepsilon \langle f, \psi_{j,k}^\varepsilon \rangle \psi_{j,k}^\varepsilon,$$

where for each $j \geqslant 1$, the set

$$\left\{ \psi_{j,k}^e = 2^{dj/2} \psi^\varepsilon(2^j x - k) : \varepsilon \in E, \ k \in \mathbb{Z}^d \right\}$$

forms a basis of $W_j$ for some wavelet $\psi$, with $E := \{0,1\}^d \setminus \{0\}$. For simplicity, we include the index $\varepsilon$ in the multi-index $k$. Finally, the set $\{\phi_{j_0,k}, \psi_{j,k}\}$ constitutes a *wavelet basis*.
For our results we will not be needing a particular wavelet basis, but any that satisfies the following key properties.

**Definition 2** (S-regular wavelet basis). *Let $S \in \mathbb{N}^*$ and $j_0 = 0$. The multiresolution wavelet basis*

$$\{\phi_k = \phi(\cdot - k), \psi_{j,k} = 2^{jd/2} \psi(2^d \cdot -k)\}$$

*of $L^2(\mathbb{R}^d)$ with associated projection kernel $K(x,y) = \sum_k \phi_k(x) \phi_k(y)$ is said to be S-regular if the following conditions are satisfied:*

*(D.1)* ***Vanishing moments and normalization:***

$$\int_{\mathbb{R}^d} \psi(x)\, x^\alpha\, dx = 0 \quad \textit{for all multi-indices } \alpha \textit{ with } |\alpha| < S, \qquad \int_{\mathbb{R}^d} \phi(x)\, dx = 1.$$

*Moreover, for all $v \in \mathbb{R}^d$ and $\alpha$ with $1 \leqslant |\alpha| < S$,*

$$\int_{\mathbb{R}^d} K(v, v+u)\, du = 1, \quad \int_{\mathbb{R}^d} K(v, v+u)\, u^\alpha\, du = 0.$$

*(D.2)* ***Bounded basis sums:***

$$M_\phi := \sup_{x \in \mathbb{R}^d} \sum_k |\phi(x-k)| < \infty, \qquad M_\psi := \sup_{x \in \mathbb{R}^d} \sum_k |\psi(x-k)| < \infty.$$

*(D.3)* ***Kernel decay:*** *For $\kappa(x,y)$ equal to $K(x,y)$ or $\sum_k \psi(x-k)\psi(y-k)$, there exist constants $c_1, c_2 > 0$ and a bounded integrable function $\phi : [0, \infty) \to \mathbb{R}$ such that*

$$\sup_{v \in \mathbb{R}^d} |\kappa(v, v-u)| \leqslant c_1 \phi(c_2 \|u\|), \qquad C_S := \int_{\mathbb{R}^d} \|u\|^S \phi(\|u\|)\, du < \infty.$$

**Case of a bounded compact $\mathcal{X} \subset \mathbb{R}^d$.** The above definition applies to wavelet systems on $\mathbb{R}^d$, but can be extended to compact domains $\mathcal{X} \subset \mathbb{R}^d$ using standard boundary-corrected or periodized constructions. Notable examples include the compactly supported orthonormal wavelets of Daubechies [11, Chapter 7] and the biorthogonal, symmetric, and highly regular wavelet bases of Cohen et al. [9]. Just as in the case of $\mathbb{R}^d$, we can build a tensor-product wavelet basis $\{\phi_k, \psi_{j,k}\}$, for example using periodic or boundary-corrected Daubechies wavelets. At the $j$-th level, there are now $O(2^{jd})$ wavelets $\psi_{j,k}$, which we index by $k \in \Lambda_j$, the set of indices corresponding to wavelets at level $j$. This coincides with the expansion used in Equation (2).

**Control of wavelet coefficients and characterization of Hölder spaces.** Remarkably, the norm of the space $\mathscr{C}^s(\mathcal{X})$ has a useful characterisation by wavelet bases - see [31] or [20] for a review on the characterisation of smoothness according to wavelet basis.

**Proposition 1.** *Let $s > 0$ we thus have the following:*

$$f \in \mathscr{C}^s(\mathcal{X}) \implies \sup_k |\langle f, \psi_{j,k} \rangle| \leqslant C|f|_s 2^{-j(s+d/2)}, \tag{64}$$

*where $C = C(\psi, S)$ is some constant that depends only on the ($S$-regular) wavelet basis.*

**Proof.** Let $\psi$ be a compactly supported wavelet in $\mathbb{R}^d$ with $S$ vanishing moments, i.e.,

$$\int_{\mathbb{R}^d} x^\beta \psi(x)\, dx = 0 \quad \text{for all multi-indices } \beta \text{ with } |\beta| < S.$$

Assume that $f \in \mathscr{C}^s(\mathbb{R}^d)$ for some $s > 0$, with $s < S$, so the wavelet vanishing moments match the regularity of $f$. Let $\psi_{j,k}(x) := 2^{jd/2}\psi(2^j x - k)$ be the wavelet at scale $j$ and location $k \in \mathbb{Z}^d$. The wavelet coefficient is given by

$$c_{j,k} := \langle f, \psi_{j,k} \rangle = \int_{\mathbb{R}^d} f(x)\psi_{j,k}(x)\, dx.$$

We define the center of the wavelet support as $x_{j,k} := 2^{-j}k$ and write a Taylor expansion of $f$ at $x_{j,k}$:

$$f(x) = P_{x_{j,k}}(x) + R_{x_{j,k}}(x),$$

where $P_{x_{j,k}}$ is the Taylor polynomial of degree $\lfloor s \rfloor$ and $|R_{x_{j,k}}(x)| \leqslant |f|_s \|x - x_{j,k}\|_\infty^s$ for $x$ near $x_{j,k}$ and where $|f|_s = \sup_{|m|=\lfloor s \rfloor} \|D^m f\|_{s-|m|}$.

Using the vanishing moments of $\psi$, we have

$$c_{j,k} = \int_{\mathbb{R}^d} R_{x_{j,k}}(x)\psi_{j,k}(x)\, dx.$$

Now perform the change of variables $u = 2^j x - k$, so $x = 2^{-j}u + x_{j,k}$ and $dx = 2^{-jd}du$:

$$c_{j,k} = 2^{-jd/2} \int_{\mathbb{R}^d} R_{x_{j,k}}(x_{j,k} + 2^{-j}u)\psi(u)\, du.$$

By the Hölder remainder estimate, we have

$$|R_{x_{j,k}}(x_{j,k} + 2^{-j}u)| \leqslant |f|_s \|2^{-j}u\|^s = |f|_s 2^{-js}\|u\|^s.$$

Therefore,

$$|c_{j,k}| \leqslant |f|_s 2^{-j(s+d/2)} \int_{\mathbb{R}^d} |\psi(u)|\|u\|^s\, du,$$

and since $\psi$ is compactly supported and smooth, the integral is finite. Hence, defining $C(\psi, s) = \int_{\mathbb{R}^d} |\psi(u)|\|u\|^s\, du < \infty$ we get the result. $\qquad\square$

**Remark.** The smoothness $s$ of $f$ translates into faster decay of the coefficients given sufficiently ($S > s$) regular wavelets.

## F Summary of the results and comparison to the literature

**Table 2:** Regret rates, parameter requirements and time complexity for online regression algorithm with $(\ell_t)$ square losses and $s > d/p$.

| Paper | Setting | Input Parameters | Regret Rate | Complexity |
|---|---|---|---|---|
| Vovk [39] | $f \in B^s_{pq},\ p,q \geqslant 1$ | $s, p, B \geqslant \|f\|_\infty$ | $T^{1-\frac{s}{s+d}}$ | $\exp(T) + Td$ |
| Vovk [40] | $f \in B^s_{pq},\ p \geqslant 2,\ q \in [\frac{p}{p-1}, p]$
$f \in \mathscr{C}^s,\ p = \infty,\ s \geqslant \frac{d}{2}$ | $s, p, B \geqslant \|f\|_\infty$ | $T^{1-\frac{1}{p}}$
$T^{1-\frac{s}{d}+\varepsilon}$ | Not feasible |
| Gaillard and Gerchinovitz [17] | $f \in W^s_p,\ p \geqslant 2,\ s \geqslant \frac{d}{2}$
$f \in W^s_p,\ p > 2,\ s < \frac{d}{2}$
$f \in \mathscr{C}^s,\ p = \infty,\ d = 1,\ s > \frac{1}{2}$ | $s, p, B \geqslant \|f\|_\infty$ | $T^{1-\frac{2s}{2s+d}}$
$T^{1-\frac{s}{d}}$
$T^{1-\frac{2s}{2s+1}}$ | $\exp(dT)$
$\exp(dT)$
$\text{poly}(T)$ |
| Liautaud et al. [29] | $f \in \mathscr{C}^s,\ p = \infty,\ s \in (1/2, 1],\ d = 1$ | $B \geqslant \|f\|_\infty$ | $T^{1-\frac{2s}{2s+1}}$ | $\text{poly}(T)$ |
| Zadorozhnyi et al. [42] | $f \in W^s_p,\ p \geqslant 2,\ s \geqslant \frac{d}{2}$
$f \in W^s_p,\ p > 2,\ s < \frac{d}{2}$ | $s, p$ | $T^{1-\frac{2s}{2s+d}+\varepsilon}$
$T^{1-\frac{s}{d}\frac{p-d/s}{p-2}+\varepsilon}$ | $\text{poly}(T)d$ |
| **This work** — Alg. 1 | $f \in B^s_{pq},\ p,q \geqslant 1,\ s \geqslant \frac{d}{2}$ or $p \leqslant 2,\ s < \frac{d}{2}$
$f \in B^s_{pq},\ p > 2,\ q \geqslant 1,\ s < \frac{d}{2}$ | $S \geqslant s,\ \varepsilon < s - \frac{d}{p}$ | $\sqrt{T}$
$T^{1-\frac{s}{d}}$ | $\text{poly}(T)S^d$ |
| **This work** — Alg. 2 | $f \in B^s_{pq},\ p,q \geqslant 1,\ s \geqslant \frac{d}{2}$ or $p \leqslant 2,\ s < \frac{d}{2}$
$f \in B^s_{pq},\ p > 2,\ q \geqslant 1,\ s < \frac{d}{2}$ | $S \geqslant s,\ \varepsilon < s - \frac{d}{p},\ B \geqslant \|f\|_\infty$ | $T^{1-\frac{2s}{2s+d}}$
$T^{1-\frac{s}{d}}$ | $\text{poly}(T)S^d$ |
| *Minimax rates* Rakhlin and Sridharan [34, 35] | $f \in B^s_{pq},\ p,q \geqslant 1,\ s \geqslant \frac{d}{2}$
$f \in B^s_{pq},\ p > 2,\ q \geqslant 1,\ s < \frac{d}{2}$ | Non constructive | $T^{1-\frac{2s}{2s+d}}$
$T^{1-\frac{s}{d}}$ | Non constructive |

**Comparison to Vovk [40].** Vovk [40] provide a general analysis for prediction in Banach spaces, focusing on the regime $s > d/p$. They achieve regret rates of $O(T^{1-1/p})$ for certain Besov spaces $B^s_{pq}$ with $p \geqslant 2$ and $q \in [p/(p-1), p]$. These rates are independent of the smoothness parameter $s$, except in the case $p = \infty$, where they obtain $O(T^{1-\frac{s}{d}})$. However, this remains suboptimal in their setting with square loss. In contrast, our analysis yields the minimax-optimal rate $O(T^{1-\frac{2s}{2s+d}})$ over a broader class of Besov spaces $B^s_{pq}$ with arbitrary $p, q \in [1, \infty]$ and $s > d/p$.

**Comparison to Vovk [39].** Vovk [39] investigates prediction under general metric entropy conditions, proposing algorithms that compete with a reference class of functions in terms of covering numbers. While their approach is highly general and applies to a broad range of normed spaces, the regret bounds they derive, of order $O(T^{1-\frac{s}{s+d}})$, still do not match the minimax-optimal rates known for functions in $B^s_{pq}$.

**Comparison to Zadorozhnyi et al. [42].** Their approach focuses on Sobolev spaces $W^s_p(\mathcal{X})$ with $p \geqslant 2$ and $s > \frac{d}{p}$, and they obtain suboptimal rates, in the regime $s < \frac{d}{2}$, of $O(T^{1-\frac{s}{d} \cdot \frac{p-d/s}{p-2}+\varepsilon})$, for arbitrarily small $\varepsilon$. In comparison, our rates $O(T^{1-\frac{2s}{2s+d}})$ are minimax-optimal over a broader class of Besov spaces $B^s_{pq}$ with arbitrary $s, p, q$ satisfying $s > \frac{d}{p}$, which include the Sobolev balls considered in their work.

**Computational complexity.** Most existing work in online nonparametric regression over Besov spaces (including Sobolev spaces), such as Rakhlin and Sridharan [34, 35], Vovk [39, 40], does not provide efficient (i.e., polynomial-time) algorithms. The work by Rakhlin and Sridharan [34, 35] offers a minimax-optimal analysis, but does not yield constructive procedures - computing the offset Rademacher complexity, as required by their method, is numerically infeasible in practice. The approach of using the Exponentiated Weighted Average (EWA) algorithm in nonparametric settings, as proposed by Vovk [39], suffers from both suboptimal regret rates and prohibitive computational complexity, since it requires updating the weights of each expert in a covering net, leading to a total cost of $O(\exp(T))$. Vovk [40] introduce the defensive forecasting approach, which also avoids efficient implementation as it relies on the so-called Banach feature map - a representation that is typically inaccessible or intractable in practice. The Chaining EWA forecaster of Gaillard and Gerchinovitz [17] achieves optimal regret bounds in the online nonparametric setting. However, its algorithm is provably polynomial-time only in the case $p = \infty$ and $d = 1$; in general dimensions and $p$, its direct implementation requires $O(\exp(dT))$ operations. Zadorozhnyi et al. [42] propose an efficient algorithm with total computational complexity of order $O(T^3 + dT^2)$. We note that their algorithm has a linear cost in $d$, making it particularly suitable for high-dimensional settings with smooth competitors in $W^s_p(\mathcal{X})$ (with $s > \frac{d}{2}$).

Finally, our algorithms are both optimal and efficient, with computational costs (after $T$ rounds) of

$$O(T \times J \times S^d) = O\left(T\frac{S}{d\varepsilon}\log_2(T)S^d\right) \quad \text{and} \quad O(T \times |\mathcal{A}|^\lambda \times J_0 \times J \times S^d) = O\left(T^{1+\frac{\lambda}{2}}\frac{S^2}{d^2\varepsilon^2}\log_2(T)^2 S^d\right)$$

for Algorithm 1 and Algorithm 2 respectively (taking a partitioning tree of maximum depth $J_0 = \lceil\frac{S}{d\varepsilon}\log_2 T\rceil$).

## G  Besov embeddings in usual functional spaces

We refer to [8, 13, 20] for precise statements of the classical embedding theorems. For convenience, we recall some of the most useful embeddings in Table 3.

**Table 3:** Classical embeddings of Besov spaces $B_{pq}^s$

| Condition on $(s, p, q)$ | Target Space | Embedding Type |
|---|---|---|
| $s > \frac{d}{p}$ | $L^\infty$ | Continuous embedding |
| $s = \frac{d}{p}, q = 1$ | $L^\infty$ | Critical embedding |
| $s > d\left(\frac{1}{p} - \frac{1}{r}\right), p < r$ | $L^r$ | Continuous embedding |
| $s_1 > s_2$ | $B_{pq}^{s_2}$ | Continuous embedding |
| $s = s, p_1 \leqslant p_2, q_1 \leqslant q_2$ | $B_{p_2 q_2}^s$ | Continuous embedding |
| $B_{pp}^s$ | $W_p^s$ | Equivalence (for $s \in \mathbb{N}$) |
| $B_{\infty\infty}^s$ | $\mathscr{C}^s$ | Norm equivalence with Hölder |

## H  Summary of optimal regret in Online Nonparametric Regression

This section summarizes the results in [34, 35] for mimimax-optimal rate of regret in the adversarial online nonparametric regression setting.

**Proposition 2** ([35])**.** *Assume the sequential entropy at scale $\varepsilon > 0$ is $O(\varepsilon^{-\alpha}), \alpha > 0$ for the target class function. Optimal regret is then summarized in the table:*

**Table 4:** Optimal regret for different loss functions

| Loss Function | Range on $\alpha$ | Optimal Regret |
|---|---|---|
| **Absolute loss** | $\alpha \in (0, 2]$ | $T^{\frac{1}{2}}$ |
| | $\alpha > 2$ | $T^{1-\frac{1}{\alpha}}$ |
| **Square loss** | $\alpha \in (0, 2]$ | $T^{1-\frac{2}{2+\alpha}}$ |
| | $\alpha > 2$ | $T^{1-\frac{1}{\alpha}}$ |

*In particular, for Hölder functions $\mathscr{C}^s(\mathcal{X}), s > 0$, and $B_{pq}^s(\mathcal{X}), s > \frac{d}{p}$ one has $\alpha = \frac{d}{s}$.*

