# OpenReview forum: "Minimax Adaptive Online Nonparametric Regression over Besov spaces"
_NeurIPS.cc/2025/Conference — NeurIPS 2025 spotlight_

### Official Review · Reviewer_eiQR · 2025-06-25

**Clarity:** 2
**Significance:** 3
**Originality:** 3
**Rating:** 5
**Confidence:** 4

**Summary:**

This paper tackles online learning in Besov space using a wavelet decomposition approach. It presents two online learning algorithms. The first can be thought of as a regression problem over a (truncated) wavelet basis containing 1 "coarse" subbasis and J-1 "detail" subbases. Each wavelet function corresponds to a cubic volume of the bounded domain built based on a doubling approach. The second algorithm builds upon this first to tackle heterogeneously smooth functions. The aforementioned cubic volumes are arranged into a tree so that predictors can be localized. They are then combined into a final prediction using specific weights, learnt jointly with the local predictors. Regret bounds and convergence results are derived for both algorithms, which are shown to adapt to the underlying target function.

**Questions:**

See "Strengths And Weaknesses"

**Ethical Concerns:**

["NO or VERY MINOR ethics concerns only"]

**Final Justification:**

I am satisfied with the authors' arguments. I, however, strongly encourage them to include these relevant discussions in the manuscript as I think it will only improve its clarity.

**Limitations:**

Yes, the authors have discussed limitations of the work in the conclusion.

**Quality:**

3

**Strengths And Weaknesses:**

The paper puts forward and analyzes minimax optimal algorithms for online approximation of Besov functions, including for functions with heterogeneous degrees of smoothness. While mostly clearly written, the density of the material in some parts warrants some improvements in clarity. In particular,

- While the conclusion mentions the exponential dependence on the smoothness parameter, it would be good to also have a discussion on the space complexity of the algorithms on the dimension. This is admittedly an issue in many wavelet methods, but I wonder if some form of adaptive tesselation could help (i.e., using an adapted wavelet basis)?

- It should be pointed out that a large portion of the complexity of the proposed algorithms is encapsulated in Assumption 1. In particular, the complexity of mirror descent updates is not taken into account in Table 2 (Appendix F). While they are polynomial in nature, the constants could be detrimental depending on the conditioning of the problem (potentially bringing another issue aside from the exponential dependence in smoothness).

- Assumption 2 is also not easy to interpret and could use more contextualization to make the paper more self-contained. As it is, the algorithms presented in the paper are really more "meta-algorithms."

- This is particularly the case for Algorithm 2, which requires substantial work from the reader to parse. In particular with respect to the description of the local predictor and its relationship with the global predictor at scale $j_0$. For instance, line 284 states that the local expert predictor (at scale $j_0$) is the restriction of the global predictor (at scale $j_0$) to the chosen subcube with the additional modification that the scaling coefficients $\alpha_{j0}$ are set to node specific coefficients $a_n$, which are not particularly clear (except for the fact that they "are supported on a grid $\mathcal{A}$ of precision $T^{-1/2}$"). It is unclear at this point how (or whether) the basis coefficients are shared among predictors (or what the role of $a_n$ is in this context). It would also be helpful to more explicitly define other quantities fundamental to the procedure, such as the active index set $\Gamma_{e,t}$.

---

> ### Author Rebuttal · Authors · 2025-07-28
>
> Dear Reviewer,
>
> Thank you very much for your constructive feedback. We appreciate your thoughtful reading and your suggestions, which helped us identify several areas for improvement. Below, we respond to your main points about complexity and assumptions.
>
> - **On exponential dependence in dimension and adaptive tessellation:**
> You are absolutely right that wavelet-based methods typically incur exponential complexity in the dimension - a known limitation of multiscale constructions. While our theoretical guarantees apply in arbitrary dimension, the practical space complexity grows as $O(S^d)$, where $S$ is the smoothness parameter and related to the size of the support of the wavelets.
> This dependence arises even with compactly supported wavelets such as Daubechies, which are known to have minimal support $O(S)$ and overlap while achieving a target regularity $S$. As such, to the best of our knowledge, this overhead is intrinsic to wavelet representations in high dimensions.
> We agree that exploring adaptive tessellation or basis/regularity adaptation is a promising direction. We would be happy to integrate any references or ideas you may have along those lines.
>
> - **Complexity of the gradient step:**
> Thank you for raising this point. While Table 2 does not explicitly break down the complexity of the update step, we clarify it now. Indeed, the 'parameter-free' descent subroutines used in Algorithm 1 are lighter then general mirror descent algorithms. In our setting, the updates are simple, closed-form computations that resemble standard gradient descent steps - see, e.g., [30, 31]. In particular, each wavelet coefficient $c_{j,k,t} \in \mathbb{R}$ is updated with a closed-form function of the scalar gradient $g_{j,k,t}$. This results in a computational cost of $O(1)$ per coefficient (simple algebraic computations), making the overall update efficient even though Assumption 1 is stated in a general form. We will clarify this point explicitly in the final version.
>
>
> - **On Assumption 2:**
> As with Assumption 1, we chose to state Assumption 2 in a general form to allow flexibility in the choice of expert subroutines. Any second-order regret algorithm satisfying this assumption - such as those in [16, 40] - can be plugged in. This abstraction allows us to present and analyze our algorithms as meta-algorithms, emphasizing their generality and modular design. That said, we agree that more context could help, and so we will consider clarifying this point in the final version.
>
> - **About Algorithm 2:**
> The basic idea behind Algorithm 2 is to best track the *local profile* of the comparator function. This is achieved by running multiple predictors $\hat f_{j_0,n}$ in parallel, each associated with a different starting scale $j_0$ and a different spatial localization $\mathcal X_n\subset \mathcal X$ - leveraging the localized support of wavelets. The predictors are aggregated using a standard second-order expert aggregation procedure (satisfying Assumption 2). More precisely, each predictor is also associated to scaling coefficients and these latter are learnt on a discretized grid $\mathcal A$ and are directly optimized through the expert aggregation process (while the detail coefficients are still learnt with Algorithm 1).
> We agree that the current description may be too dense, and we will revise the final version to better explain this mechanism.

---

> > ### Comment · Reviewer_eiQR · 2025-08-05
> >
> > I thank the authors for their response. I am satisfied with their arguments (as I already was). I, however, strongly encourage them to include these relevant discussions in the manuscript as I think it will only improve its clarity.

---

> > > ### Author Response · Authors · 2025-08-08
> > >
> > > We thank the reviewer and will incorporate these discussions in the final version.

---

### Official Review · Reviewer_LkUW · 2025-06-27

**Clarity:** 2
**Significance:** 2
**Originality:** 2
**Rating:** 4
**Confidence:** 3

**Summary:**

This paper studies the problem of online nonparametric regression where the hypothesis class is a general Besov space. An adaptive algorithm based on wavelet decomposition and parameter free online learning is first proposed to achieve the minimax optimal regret bound without knowing the parameters of the hypothesis class beforehand. Then, inspired by (Liautaud et al., 2025), an improved algorithm based on expert aggregation further improves the result to local adaptivity.

**Questions:**

1. I did not understand the expert construction in Algorithm 2. Could the authors elaborate on it? The description of the algorithm is also dense, therefore if possible, I would suggest revising the writing to highlight the key idea.

2. I am also wondering if Algorithm 1 can directly achieve local adaptivity without expert aggregation, or put in another way, whether the analysis of Theorem 1 is loose. Any comment or insight would be helpful.

**Ethical Concerns:**

["NO or VERY MINOR ethics concerns only"]

**Final Justification:**

Overall, my evaluation of the paper is that it makes some good contributions in terms of the final results, but the significance, novelty and impact of these results are not particularly strong. It probably started as an attempt to generalize the work of Liautaud et al, with the use of wavelet machinery being the main technical novelty. However, the works of Baby & Wang and Zhang et al, which the authors were not aware of, have shown similar ideas before, therefore the significance and novelty of this paper are unfortunately weakened to a certain extent.

I have no major problem seeing this paper accepted, but if there exists an acceptance threshold, then I feel this paper is not significantly above it. As the result, I've increased my final score to 4.

**Limitations:**

Yes

**Paper Formatting Concerns:**

No formatting issues.

**Quality:**

3

**Strengths And Weaknesses:**

Strengths

- The considered problem is a challenging one that meaningfully extend the setting from an earlier work (Liautaud et al., 2025) with a similar theme.

- The technical development in this paper is solid as far as I can tell.

- The results are quantitatively strong, achieving optimality across a number of settings and parameter regimes.

Weaknesses

- It is claimed in the paper that the proposed algorithm is the first adaptive algorithm to bridge wavelet theory with online nonparametric learning, which I think is not accurate. Online nonparametric regression is known to be related to the so called dynamic online learning problem as discussed in the related work section of Baby and Wang (2019), where there is no context but the comparing hypothesis can be time varying. This setting is a special case of online nonparametric regression where the feature space is just the sequence of time indices 1,...,T. Baby and Wang's works (2019,2020) combined wavelet theory with averaging and the VAW forecaster to achieve the optimal dynamic regret, extending earlier works on trend filtering to the sequential setting. More recently, Zhang et al (2023) combined wavelet with parameter free online learning to tackle the case with general convex loss functions, and their algorithm appears to be identical to Algorithm 1 proposed by the submission despite using discrete wavelets. The relation to these works needs to be thoroughly discussed.

- The technical development of the paper closely parallels the earlier work of Liautaud et al (2025), and the novelty compared to that is not sufficiently highlighted. Quantitatively, the proposed algorithm can handle more general hypothesis classes than Liautaud et al's, but given the similarity of the two algorithms, it is currently less clear what is the key technical contribution and how significant is that.

References

Baby, Dheeraj, and Yu-Xiang Wang. "Online forecasting of total-variation-bounded sequences." Advances in Neural Information Processing Systems 32 (2019).

Baby, Dheeraj, and Yu-Xiang Wang. "Adaptive online estimation of piecewise polynomial trends." Advances in Neural Information Processing Systems 33 (2020): 20462-20472.

Zhang, Zhiyu, Ashok Cutkosky, and Yannis Paschalidis. "Unconstrained dynamic regret via sparse coding." Advances in Neural Information Processing Systems 36 (2023): 74636-74670.

---

> ### Author Rebuttal · Authors · 2025-07-28
>
> Dear Reviewer,
>
> Thank you for your feedback and for pointing out these references. Below, we address your comments point by point and hope these clarifications help demonstrate the broader scope and novelty of our approach.
>
> **I) On prior work of Baby and Wang:**
> We acknowledge that wavelet-based techniques have been explored in the line of work by Baby and Wang. Nevertheless, their analysis focuses on a specific application: they use an online algorithm (VAW) for batch denoising with wavelets, where the data is assumed to be i.i.d. and lies on a uniform grid. As such, their setting is stochastic and non-adversarial, and aligns closely with problems studied in the signal processing community. In contrast, our work develops a wavelet-based algorithm specifically for the *online adversarial learning* setting. We will make this distinction more precise in the related work section of the final version.
>
> That said, our setting and contributions significantly generalize this line of works in several key aspects, here are some:
> - while their method apply to specific discrete Sobolev- or Hölder-type classes, they remain confined to structured sequences over 1D grids. The input domain is implicitly the time axis $\{1, \dots, T\}$. In contrast, our algorithms operate in a general contextual setting $x_t \in \mathbb{R}^d$, without relying on grid structure or discretization. This allows approximation over general domains $\mathcal X$ via multiscale wavelet decompositions and in an adversarial setting. They use the discrete wavelet transform (DWT) over sequences, while we construct and analyze algorithms using continuous wavelet expansions in general Besov spaces $B^s_{p,q}$.
> - they consider discrete sequences $\theta\_{1:T} = [f(x_t)]\_{1:T} \in \mathbb{R}^T$ over the 1D time index $\{1, \dots, T\}$, with regularity enforced via finite differences (e.g., total variation of order $k, \mathrm{TV}^k(C_T)$). Typically, in our setting the data context $(x_t)$ does not live on a grid and is unknown in advance. It is sequentially revealed in an arbitrary ordering. In our setting, their total variation constant $C_T$ could behave as $C_T = O(T)$, making their regret upper bounds vacuous.
> - finally, their function class $\mathrm{TV}^k(C_T)$ is a particular class of functions sandwiched between Besov-type spaces
>   $
>   B^{k+1}\_{11} \subset \mathrm{TV}^k \subsetneq B^{k+1}\_{1\infty}
>   $
>   all of which are naturally handled by our method, since we are competitive with any $B_{pq}^s$. In particular our algorithm also achieves their minimax optimal dynamic regret in their setting. Letting $\ell\_t(\hat y) = (y_t - \hat y)^2$, suppose $f(x\_t) = \theta\_{1:T}[t]$ for some $\theta\_{1:T} \in \mathrm{TV}^k(C\_T) \subset B^{k+1}\_{1,\infty}$, and $y_t = \theta_{1:T}[t] + \varepsilon_t$ with independent Gaussian noise.
> While our predictions $(\hat f_t)$ make *no stochastic assumptions* beyond the boundedness of $(y_t)$, our regret (for $s = k+1$) directly upper bounds their dynamic regret when taking the expectation (see their discussion in Appendix B, paragraph competitive online nonparametric regression). Thereby, our method also achieves their optimal dynamic regret of $O(T^{1/(2k+3)})$.
>
> We will add these distinctions in the related of the final version.
>
> **II) About Zhang et al. (2023):**
> The recent work by Zhang et al. (2023) introduces an online algorithm that combines discrete wavelets with parameter-free learning to minimize dynamic regret under general convex losses. While their method does share structural similarities with our Algorithm 1 - both maintain and update coefficients using a parameter-free subroutine - the wavelet basis they use, their goals, and their analysis differ fundamentally from ours.
> Zhang et al. focus on obtaining sparse dynamic regret bounds in a parametric framework (finite dictionary). Although they draw connections to nonparametric regression, by considering large dictionaries thanks to their sparsity results, they do not derive explicit rates with respect to function classes such as Besov or Hölder spaces. In contrast, selecting an appropriate wavelet basis and analyzing rates with respect to these function classes is central to our approach - and not addressed in their analysis.
> Moreover, their algorithm does not consider the exp-concave setting, where, similar to our Algorithm 1, it would fail to achieve minimax rates. In contrast, our Algorithm 2 is specifically designed for this setting and introduces both conceptual and technical innovations to achieve optimal performance.
> Thank you for pointing out this relevant reference; we will add it to the related work section and clarify this distinction in the final version.
>
> **III) Contributions and clarification regarding Algorithm 2:**
>
> - *About contributions in regard to the work of Liautaud et al.:*
> Our current work significantly generalizes the previous setting by competing against a much broader class of functions - namely, Besov-smooth functions - whereas the earlier work of Liautaud et al. focused on Lipschitz-smooth functions. This extension requires a substantially more intricate analysis of the algorithm due to the richer structure and potential irregularity of Besov spaces. For instance, the need for a nonlinear oracle analysis and sparse wavelet representations across scales, or the challenge of handling spatially inhomogeneous smoothness. In particular, the analysis must adapt to both the regularity and integrability properties encoded by the Besov parameters $(s, p, q)$, which introduces significant technical complexity compared to standard Hölder classes with $p=q=\infty$.
> To our knowledge, this is the first algorithm designed and analyzed with optimal guarantees against adversarial Besov-smooth environments, covering the full range of smoothness regimes embedded in $L^\infty$, in a fully *online* and *adversarial* setting. In addition, we provide concrete examples to show that our method achieves faster rates via local adaptation, particularly when the target function exhibits high regularity locally. In contrast, the approach in Liautaud et al. is inherently limited to $s \leq 1$.
> We will consider adding further intuition and key steps from our analysis to the main text in order to better highlight these contributions.
>
> - *About Algorithm 2:* It introduces a spatially adaptive, tree-structured framework that enables local adaptation to heterogeneous Besov regularity, improving over the global guarantees of Algorithm 1. Briefly, it leverages the localized support of wavelets to build region-specific predictors with tailored coefficients, which are then aggregated using a second-order expert procedure to achieve local adaptivity and optimal regret. In particular, it supports recovery in settings with spatially varying smoothness $s$, adapting *locally* instead of paying for the lowest regularity *globally*.
> Regarding your question: Algorithm 1 does not achieve such local adaptivity, as it applies a uniform strategy across the entire domain $\mathcal X$. This limitation is intrinsic to the global regret bound in Theorem 1. Algorithm 2 overcomes this by aggregating local predictors specialized to subregions and tuned to the corresponding local regularity levels - thereby recovering different local rates within a unified framework. We also highlight that Algorithm 2 supports acceleration for exp-concave losses, which is not captured by Algorithm 1.
> Finally, we agree that the expert construction in Algorithm 2 may appear dense, and we will work to clarify its core idea in the main of the final version.

---

> > ### Comment · Reviewer_LkUW · 2025-08-03
> >
> > First, thank you for the explanation.
> >
> > It's claimed in the rebuttal that the work of Baby and Wang '19 assumes iid stochastic data, which is not correct. They studied the setting of agnostic adversarial loss plus stochastic noise, as shown in their protocol, Figure 1. Using the notations there, the adversarial component comes from $\theta$, and the stochastic part comes from $Z$ which is iid. This setting originates from Besbes et al '15, and is no easier than the noiseless adversarial setting typically studied by other online learning papers.
> >
> > The rebuttal also says the works of Baby and Wang assume the context lives on the fixed grid of {1,...,T} and is revealed isotonically. I think to some extent this is not accurate, please see a later work of theirs in AISTATS 2021, where non-isotonic ordering is considered.
> >
> > Compared to the works of Zhang et al. '23, it's claimed that a major novelty of this paper is the analysis of exp-concave losses, which I agree with.
> >
> > There are some additional questions:
> >
> > 1. I see from the rebuttal that using the continuous wavelet transform instead of the discrete one is a major novelty of this paper compared to the related works I mentioned. Could you please explain the reason? Naively I tend to think their main ideas are the same, and the differences are possibly well-studied in wavelet theory which is arguably not so relevant to online learning.
> >
> > 2. If I understood it correctly, it's claimed that Algorithm 1 fundamentally cannot achieve local adaptivity, and this is due to the algorithm design rather than the analysis. Algorithm 2 improves that by aggregation, or model selection. Again, a naive thought: Algorithm 1 is based on parameter-free online learning, and a somewhat well-known capability of parameter-free online learning is that it can automatically perform model selection, see the work of Cutkosky (2019). Zhang et al (2023) also touches upon this. So I wonder whether this intrinsic model selection capability of Algorithm 1 is strong enough, so that the explicit aggregation in Algorithm 2 might not be required.
> >
> > 3. Given all the discussion so far, I'm a little lost about the exact contribution and novelty of this paper. It seems that narrowly, compared to each individual paper of Baby and Wang, Zhang et al and Liautaud et al, this paper has nontrivial new components, but these new components also exist in another existing paper of these authors. So I'd like to ask for another concise summary of the key novel idea, and a justification of its significance.
> >
> > Omar Besbes, Yonatan Gur, and Assaf Zeevi. Non-stationary stochastic optimization. Operations research, 63(5):1227–1244, 2015.
> >
> > Baby, Dheeraj, Xuandong Zhao, and Yu-Xiang Wang. "An optimal reduction of tv-denoising to adaptive online learning." In International Conference on Artificial Intelligence and Statistics, pp. 2899-2907. PMLR, 2021.
> >
> > Cutkosky, Ashok. "Combining online learning guarantees." In Conference on Learning Theory, pp. 895-913. PMLR, 2019.

---

> ### Author Response · Authors · 2025-08-04
>
> We thank the reviewer for engaging in the discussion. Below, we respond to their comments and hope to clarify any misunderstandings.
>
> **Stochastic assumption in Baby&Wang (2019)**
>
> Although $\theta_i$ is described as adversarial in their Figure~1, their setting corresponds to a stochastic approximation of a fixed function $f$, as we stated. Indeed, defining a fixed function $f$ such that $f(x_i) = \theta_i$ for all $i$, their setting amounts to estimating this fixed function $f$ from noisy observations $f(x_i) + Z_i$, where the $Z_i$ are i.i.d., zero-mean, $\sigma^2$-subGaussian noise with *known* variance parameter $\sigma$. This interpretation is made explicit in their follow-up work published at AISTATS 2021. Moreover, in their setting, each input $x_i$ is unique and queried only once, with the corresponding observation $\theta_i + Z_i$. This allows them to define different values $\theta_i$ for each $i$. If the same point $x_j = x_i$ were queried multiple times (which they do not consider), their analysis would require to assume $\theta_j = \theta_i$.
>
> Crucially, their setting does not cover fully adversarial sequences $\theta_i$, since all their results depend on the total variation $C_n = \sum_i |\theta_i - \theta_{i+1}|$, which only makes sense if the sequence is generated from some underlying smooth function $f$, so that $C_n$ remains small. Applying their bounds to arbitrary adversarial sequences $(\theta_i)$ could lead to $C_n$ of order $T$, resulting in linear regret and vacuous guarantees. Furthermore, their analysis heavily relies on the assumption of i.i.d., zero-mean, $\sigma^2$-subGaussian noise with known $\sigma$, particularly for breakpoint detection in the sequence $\theta_i$, via concentration arguments. These techniques would not extend to a fully adversarial framework.
>
> In contrast, all our results hold for *any* adversarial design of the data (inputs and observations), while still achieving sublinear optimal regret. The smoothness assumption is imposed only on the comparator function---not on the observations, nor on the inputs.
>
> **Isotonic assumption in Baby and Wang (Aistats, 2021)**
>
> First, note that Baby and Wang (2021) also operates within the stochastic framework described above, which is significantly easier than ours, as it falls under the well-specified setting where the data are generated from a fixed smooth function $f$ with additive i.i.d. noise. By contrast, the purely adversarial regime we consider is more challenging due to the bias it introduces in the analysis. For instance, in their Lemma 7, they exploit the decomposition
> $$
> 	\mathbb{E}\left[(y_i - A_I(j))^2 - (y_j - \theta_j)^2\right] = \mathbb{E}\left[(A_I(j) - \theta_j)^2\right],
> $$
> which crucially relies on the independence and zero-mean properties of the noise. In the adversarial setting, a cross-term of the form $2 (y_i - A_I(j))(y_j - \theta_j)$ would arise and contribute a linear term to the regret, significantly deteriorating the rate. Moreover, their analysis is specifically tailored to the squared loss, while we consider more general loss functions.
>
> Second, although we agree that Baby and Wang (2021) does allow non-isotonic orderings, it still makes two key simplifying assumptions:
>
> - The set of inputs $\{x_1,\dots,x_T\}$ is fixed and revealed to the learner in advance. Indeed, as they acknowledge in their Remark 10, their approach *fails under an adaptive adversary* unless the entire sequence of covariates $(x_1, \dots, x_T)$ is *revealed ahead of time*. This corresponds to the *transductive* setting in online learning, where better regret guarantees are known to be achievable [1]. In contrast, in our setting, the inputs are arbitrary and revealed sequentially to the learner.
> - They still assume a bounded variation condition of the form $\sum_{t=1}^{T-1} |f(x_t) - f(x_{t+1})| \leq C_n$ with $C_n = o(T)$. Such an assumption is only motivated in the isotonic stochastic approximation of a fixed smooth function $f$, and does not apply to adversarially chosen sequences.
>
> [1] Qian et al., 2024. Refined risk bounds for unbounded losses via transductive priors. arXiv:2410.21621.
>
> **Novelty with respect to Zhang et al. '23**
>
> We also show that our method achieves minimax rates in the adversarial nonparametric regression setting, which is far from obvious from Zhang et al. (2023), who only provide results with respect to finite-dimensional families of dictionaries. Indeed, by selecting an appropriate dictionary and following derivations similar to ours, we believe that the techniques of Zhang et al. (2023) could potentially recover our results for convex losses. However, they did not prove this, and we view the extension to nonparametric convex losses as a contribution in its own right.
>
> In the exp-concave setting, not only do they not provide explicit rates for the nonparametric regression setting, but we also believe that their method---like our Algorithm 1---is suboptimal in this regime.

---

> > ### Author Response · Authors · 2025-08-04
> >
> > **Q1. On the use of continuous wavelet representation**
> >
> > It enables approximation of functions over $\mathbb{R}^d$ without relying on a fixed discretization grid of the inputs.
> > For example, in the papers Baby & Wang (2019, 2021), their algorithms remain constrained to a fixed input grid revealed beforehand to the learner. In contrast, our framework operates in a **fully online** and **adversarial** setting, with covariates and labels revealed **sequentially**, and still achieves **minimax optimality**. This fundamental difference in assumptions and capabilities is typically enabled by our use of continuous wavelet representation, without grid constraints or prior access to the input sequence.
> >
> > **Q2. On the need of Algorithm 2 (local adpativity / minimax rates for expconcave losses)**
> >
> > A main limitation of Algorithm 1 arises in the exp-concave setting. Indeed, it only relies on parameter-free subroutines that adapt to the norm of the underlying parameter $c$, and incur a regret of order $O(c \sqrt{T})$. While this is suitable in the convex case, it is suboptimal for exp-concave losses, where a regret of order $O(\log T)$ can be achieved when $c$ is large. Algorithm 2 addresses this issue by selecting, at all scales simultaneously, which parameters should be estimated at a cost of $O(\log T)$ and which should be estimated at a cost of $c \log T$. This selection mechanism is key to achieving minimax optimal rates in the exp-concave setting.
> >
> > Local adaptivity also emerges as a desirable by-product of Algorithm 2. It remains unclear whether Algorithm 1 can directly yield local adaptivity in the convex case, and this is an interesting question. However, we strongly doubt that Algorithm 1 can achieve optimal rates in the exp-concave setting, due to the inherent $O(\sqrt{T})$ regret bound of parameter-free algorithms.
> >
> >
> > **Q3. We follow up on our previous message with a concise summary of our contributions**
> >
> > - We design and analyze a novel algorithm based on parameter-free updates over wavelet expansions, which minimizes regret in the fully adversarial nonparametric regression setting with respect to general *Besov spaces*. These spaces capture a wide range of function classes, including low-regularity functions. We analyze the algorithm under both convex and exp-concave loss functions. In contrast, Baby et al. (2019, 2021) consider a significantly easier setting---a stochastic approximation of a fixed function with i.i.d. noise, where the inputs are fixed and known beforehand (i.e., the transductive setting). Their framework can be seen as a special case of ours. Zhang et al. (2023) do not provide explicit rates in the nonparametric setting and do not address exp-concave losses.
> >
> > - We derive **minimax-optimal regret bounds** that adapt to the unknown smoothness $s$ and integrability parameters $p, q$ of the target function class, in both *adversarial* and *exp-concave* loss settings. Crucially, our method is *constructive*, thereby addressing an open question (see, e.g., Rakhlin 2014, 2015) of whether polynomial-time algorithms can achieve minimax regret against general Besov functions in an adversarial setting, without assuming any stochastic model. To the best of our knowledge, no previously known constructive and efficient method achieves minimax rates in this general framework.
> >
> > - We also study **local adaptation to spatially varying smoothness**. While existing global methods incur regret based on the worst-case smoothness across the domain, our approach adapts to local regularity and achieves strictly improved regret bounds in heterogeneous regimes—a setting that is both theoretically challenging and highly relevant in practice. Prior work on local adaptation (e.g., Kuzborskij et al. 2020; Liautaud et al. 2020) focused only on Lipschitz-continuous functions and their Lipschitz constant. We generalize their techniques to significantly more complex function classes such as Besov spaces.

---

> > > ### Comment · Reviewer_LkUW · 2025-08-07
> > >
> > > Thank you for the detailed follow-up.
> > >
> > > Overall, my evaluation of the paper is that it makes some good contributions in terms of the final results, but the significance, novelty and impact of these results are not particularly strong. It probably started as an attempt to generalize the work of Liautaud et al, with the use of wavelet machinery being the main technical novelty. However, the works of Baby & Wang and Zhang et al, which the authors were not aware of, have shown similar ideas before, therefore the significance and novelty of this paper are unfortunately weakened to a certain extent.
> > >
> > > I have no major problem seeing this paper accepted, but if there exists an acceptance threshold, then I feel this paper is not significantly above it. As the result, I've increased my final score to 4.

---

> > > > ### Author Response · Authors · 2025-08-08
> > > >
> > > > Thank you for your response and the preceding discussion. We will include these references with appropriate discussion and comparison.

---

### Official Review · Reviewer_ZsGV · 2025-07-06

**Clarity:** 4
**Significance:** 3
**Originality:** 3
**Rating:** 5
**Confidence:** 4

**Summary:**

This paper develops online learning algorithms that adapt to local irregularities of the function class played by the adversary. The algorithms are based on parameter free online linear optimization algorithms, applied to each component of a wavelet decomposition separately, then aggregated. The resulting bounds are adaptive to the besov norm of a comparator function. Theorem 1 gives a trichotemy showing that different rates are possible with the stated algorithms depending on the relationship between dimension and smoothness. Theorem 2 gives further bounds in the case of spatially inhomogeneous smoothness. The boudns match the minimax rates in a wide range of cases.

**Questions:**

-

**Ethical Concerns:**

["NO or VERY MINOR ethics concerns only"]

**Final Justification:**

I am still in favour of accepting. I maintain my score.

**Limitations:**

-

**Paper Formatting Concerns:**

some references cite arxiv version of papers that are published.

**Quality:**

4

**Strengths And Weaknesses:**

The paper is well written and easy to follow. The work combines recent results in parameter free optimization with sophisticated techniques from applied maths in order to develop something broadly applicable and useful, as well as of theoretical interest.

---

> ### Author Rebuttal · Authors · 2025-07-31
>
> Dear Reviewer,
>
> Thank you very much for encouraging feedback. We sincerely appreciate your positive assessment of our work. We will make sure to replace citations of arXiv versions with the corresponding published versions in the final submission.

---

### Official Review · Reviewer_GTjW · 2025-07-07

**Clarity:** 3
**Significance:** 3
**Originality:** 3
**Rating:** 5
**Confidence:** 2

**Summary:**

This paper studies online nonparametric regression using a competitive framework, i.e., bounding regret compared to the best predictor in a given function class (here, given by a Besov space). Specifically, the paper proposes a wavelet-based Algorithm 1 that is simultaneously competitive with predictors from any Besov space, up to a predetermined smoothness level $S$. The paper then proposes a "spatially adaptive" Algorithm 2, which uses a collection of experts  (each trained using Algorithm 1) assigned to different dyadic partitions of the space, and shows that Algorithm 2 obtains correspondingly stronger regret guarantees.

**Questions:**

1. What exactly is meant by the term "parameter-free" used throughout the manuscript? Does this refer to the fact that the parameters $s$, $p$, etc. need not be available to the algorithm? How does this differ from the term "adaptive" used throughout the paper and elsewhere in the statistical literature?

2. A key result in the statistical literature on Besov function estimation (due to [13, 14, etc.]) is that linear estimators (e.g., simple projection or kernel estimators) necessarily fail to achieve minimax optimal convergence rates (under certain loss functions) over many Besov spaces, and that some nonlinear mechanism, such as thresholding, is needed. If I understand correctly, the construction of the active wavelet set plays a similar role here. Is there an analogous lower bound for this setting, e.g., where a simpler class of estimators that are competitive with Holder-smooth functions provably fail to compete with Besov-smooth functions? This might also be related to the spatial adaptivity advantages of Algorithm 2 over Algorithm 1, but I'm not sure if this is the same, since usually spatial adaptivity is within a single Besov space rather than across different (locally restricted) Besov spaces.

3. The loss function is assumed to be both convex and Lipschitz; does this need to be true on the whole real line, or can this assumption be restricted to a bounded domain? On an unbounded domain, this is a lot more restrictive (i.e., the loss function is essentially linear outside some bounded region).

4. Could the authors provide a bit more intuition for Assumption 1? It's not really clear to me what it means. Maybe an example (of the type of behavior it is restricting) would be helpful?

**Ethical Concerns:**

["NO or VERY MINOR ethics concerns only"]

**Final Justification:**

Thanks to the authors for their response. I found the comparisions to the batch setting helpful, and suggest these discussions be added to the paper. I still find Assumption 1 rather mysterious and think concrete examples (violating & satisfying the assumption) would help here. Also, I am not familiar enough with the online learning literature to follow the concerns about novelty (compared to prior work by Baby & Wang and Zhang et al.) brought up by Reviewer LkUW. Given these, I have reduced by confidence from 3 to 2, but am retaining my overall rating of 5.

**Quality:**

3

**Strengths And Weaknesses:**

### Strengths
1. Though dense, the paper is generally well-written and easy to follow. In particular, the paper does a good job both discussing demonstrative special cases/examples (e.g., Figure 3) and summarizing the results at a high level (e.g., with Figure 2).
2. The proposed method does not require any strong knowledge of the data-generating process; in particular, it is impressive that the same algorithm is competitive with predictors from any Besov space with $s < S$.

### Weaknesses
1. As someone used to nonparametric regression in an offline, statistical/stochastic context and unfamiliar with the online learning setting, it took some adjustment to follow the paper's ideas, such as the "competitive" evaluation framework. The competitive framework seems designed to optimize prediction performance (under a particular loss function) while minimizing the assumptions made the data-generating process. However, it's unclear to me how much one can infer about the underlying data-generating process from this approach. For example, if we assume that the data is generated from a Besov-smooth function (i.e., $f(x) = \mathbb{E}[Y|X = x]$ for some $f \in B_{p,q}^s$) and train a predictor $\hat f$, what do the given bounds tell us about $f$ (perhaps with some additional assumptions on the noise)?
2. The main paper doesn't really discuss the proofs of the main results at all. I understand that space is limited, but a few sentences would be nice to give an idea of the key ideas/overall approach.

---

> ### Author Rebuttal · Authors · 2025-07-25
>
> Dear Reviewer,
>
> Thank you very much for your constructive and insightful comments.
>
> **I) Parameter-freeness and adaptivity:** Your question about the distinction between "parameter-free" and "adaptive" indeed points to an important subtlety in terminology, and we appreciate the opportunity to clarify this in our paper.
>
> *First about the terminology:* in the online optimization literature, the term **'parameter-free'** refers to algorithms that do not require tuning any user-specified hyperparameter (e.g., a learning rate or regularization coefficient), in order to achieve optimal regret bounds. More precisely, it typically denotes algorithms satisfying bounds of the form
>
> $$
> \sum_{t=1}^T g_t(c_t - c) \leq O\left(|c| \sqrt{\sum_{t=1}^T g_t^2}\right),
> $$
>
> for any comparator $c \in \mathbb R$. This implies *automatic adaptivity to the norm* $|c|$, without prior knowledge on it. It is a central notion in the monograph by F. Orabona (*A Modern Introduction to Online Learning*, Chapter 10), as well as in [10, 31]. This is sometimes also called 'comparator(-norm) adaptivity' (see [30]).
>
> *Regarding our setting:* this parameter-freeness property is encapsulated in a general Assumption 1 (with starting at $c_1 = 0$), which allows us to plug in any such online algorithm as a gradient step in our wavelet-based Algorithm 1. This assumption is then crucial in our analysis: we bound part of the regret using this norm-adaptive behavior with respect to each wavelet coefficient. Intuitively, it allows us to control the regret through both the structure and magnitude of the scale/detail coefficients $|c_{j,k}| \lesssim 2^{-j s'}$.
>
> *Relation to statistical adaptivity:* in contrast, **'adaptive'** in the statistical literature usually refers to algorithms that utomatically adapt to unknown regularity parameters of the target function (such as smoothness $s$, integrability $p$, etc.), often achieving minimax-optimal rates without prior knowledge of these quantities. In our case, we use *parameter-free subroutines* to build a *statistically adaptive* method: our regret bounds hold uniformly over Besov functions indexed by any $s, p, q$, without these parameters being given to the algorithm.
>
> **II) Convexity and Lipschitz assumptions on the losses:**
>
> In our analysis, the loss function only needs to be convex and Lipschitz on the domain where the algorithm operates - i.e., over the range of predictions $\hat f_t(x_t)$. Typically, if we assume that the comparator $f$ is such that $|f|_\infty \leq B$, we may clip $\hat f_t(x_t)$ (as done for Alg. 2) and require convexity and Lipschitz assumptions on $[-B,B]$ only.
>
>
> **III) Nonlinear estimation and spatial adaptivity:** Thank you for this insightful and technically deep question.
>
> - You are absolutely right that, in the offline statistical setting (Donoho et al.), linear estimators fail to achieve minimax rates over many Besov spaces. In our online setting, the same phenomenon arises, hence the need for a nonlinear mechanism. The algorithm dynamically updates the wavelet coefficients in the active set over time, thereby mimicking a nonlinear adaptation oracle.This is performed via the use of parameter-free subroutines that - roughly speaking - directly track the best (i.e. thresholded) wavelet coefficients. Indeed this mechanism is essential to achieving minimax-optimal regret bounds while adapting to the unknown Besov regularity of the target function. We will add comments in the main text to better highlight this, and introduce the nonlinear oracle that our algorithm implicitly mimics.
>
> - *Regarding your observation 'this might also be related to the spatial adaptivity advantages of Algorithm 2 over Algorithm 1':* Our Algorithm 1 already mimicks the behavior of the ideal selection of $(j,k)$ indices - that is, the optimal set of active wavelet coefficients, which serves as the indices constituting the (thresholded) nonlinear oracle. This is achieved through the use of parameter-free subroutines that effectively target the 'thresholded coefficients'.
> In contrast, Algorithm 2 also supports *local* adaptivity across different regularity classes, not just within a fixed Besov space. Donoho et al’s notion of spatial adaptivity in [13] refers to adaptation to inhomogeneities within a single (global) Besov space - e.g., detecting jumps or peaks via wavelet thresholding. Our method goes beyond the (global) spatial adaptivity considered in [13] and enables the algorithm to handle heterogeneous smoothness $s$. In particular, we show that this local strategy can adapt to different levels of smoothness across regions and improves over global guarantees. We will clarify this distinction more explicitly in the final version and add comments to emphasize the parallel with the litterature of Donoho's and al.
>
> - *About the lower bound:* While our current focus is on establishing that our algorithm achieves optimal performance across a range of Besov spaces, we do not provide an explicit lower bound showing the failure of linear estimators in the online setting. Nevertheless, such lower bounds can be derived: since the *offline/batch i.i.d.* smoothing problem is strictly 'easier' than the *online* one, any lower bound on the minimax risk (e.g., MSE) of linear smoothers in the offline setting (as in [14]) implies a lower bound on the regret of linear predictors in our online setting. We will consider adding such a result in the supplementary material of the final version to further motivate the need for a nonlinear mechanism.
>
> **IV) Guarantees in the batch setting:**
>
> Our guarantees in the paper are stated in terms of *regret* and  we make no assumptions on the data-generating process - only boundedness of the observations. However, one may naturally be interested in guarantees in a *batch* or *statistical* setting. Indeed, our *regret* upper-bounds can be used to derive *excess risk* bounds via standard *online-to-batch conversion* (e.g., see N. Cesa-Bianchi, A. Conconi, and C. Gentile. *On the generalization ability of on-line learning algorithms*). Typically, if we now consider iid data $(X,Y)$, and if we define a batch estimator $\bar f_T = \frac{1}{T} \sum_{t=1}^T \hat f_t$ as the average of the online predictors, then the expected excess risk of $\bar f_T$ with respect to any Besov-smooth function $f$ would typically scale as our regret bound divided by $T$. Note that $\mathbb E[Y|X]$ may be any bounded measurable function and non necessary Besov.
> Finally, if the question is to obtain guarantees on the mean integrated squared error (MISE), i.e., $\mathbb{E}[\|\bar f_T - f\|_2^2]$, with $f \in B^s\_{pq}$ then in the specific i.i.d. case where $Y = f(X) + \varepsilon, E[\varepsilon | X] = 0$ with i.i.d. uniform design, our regret bounds provide also optimal guarantees whenever $s \ge d/2$. Interestingly, we believe that Algorithm 2 could also be extended to derive guarantees for general $L^{p'}$-risks with $p' > p$ in our setting - a promising and challenging direction for future work, given the rich literature by Donoho, Johnstone, Kerkyacharian, and Picard.

---

### Decision · Program_Chairs · 2025-09-17

**Decision:**

Accept (spotlight)

**Comment:**

The paper proposes an adaptive wavelet based  algorithm for online  non-parametric regression in the adversarial setting that adapts to the parameters including smoothness parameters of the Besov space. The algorithm is the first constructive algorithm to obtain the minimax optimal regret bound for this setting. The reviewers all seem to be in agreement that the paper is worth an accept. I agree with them.